# Flura-seq identifies organ-specific metabolic adaptations during early metastatic colonization

**Harihar Basnet[1], Lin Tian[1], Karuna Ganesh[1,2], Yun-Han Huang[1,3,4], Danilo G Macalinao[1,4], Edi Brogi[5], Lydia WS Finley[6], Joan Massagué[1]\***

[1]Cancer Biology and Genetics Program, Sloan Kettering Institute, Memorial Sloan Kettering Cancer Center, New York, United States; [2]Department of Medicine, Sloan Kettering Institute, Memorial Sloan Kettering Cancer Center, New York, United States; [3]Weill Cornell/Rockefeller/Sloan Kettering Tri-Institutional MD-PhD Program, New York, United States; [4]Louis V. Gerstner, Jr. Graduate School of Biomedical Sciences, Memorial Sloan Kettering Cancer Center, New York, United States; [5]Department of Pathology, Memorial Sloan Kettering Cancer Center, New York, United States; [6]Cell Biology Program, Sloan Kettering Institute, Memorial Sloan Kettering Cancer Center, New York, United States

**Abstract** Metastasis-initiating cells dynamically adapt to the distinct microenvironments of different organs, but these early adaptations are poorly understood due to the limited sensitivity of in situ transcriptomics. We developed fluorouracil-labeled RNA sequencing (Flura-seq) for in situ analysis with high sensitivity. Flura-seq utilizes cytosine deaminase (CD) to convert fluorocytosine to fluorouracil, metabolically labeling nascent RNA in rare cell populations in situ for purification and sequencing. Flura-seq revealed hundreds of unique, dynamic organ-specific gene signatures depending on the microenvironment in mouse xenograft breast cancer micrometastases. Specifically, the mitochondrial electron transport Complex I, oxidative stress and counteracting antioxidant programs were induced in pulmonary micrometastases, compared to mammary tumors or brain micrometastases. We confirmed lung metastasis-specific increase in oxidative stress and upregulation of antioxidants in clinical samples, thus validating Flura-seq's utility in identifying clinically actionable microenvironmental adaptations in early metastasis. The sensitivity, robustness and economy of Flura-seq are broadly applicable beyond cancer research.
DOI: https://doi.org/10.7554/eLife.43627.001

**\*For correspondence:**
j-massague@ski.mskcc.org

## Introduction

Metastasis is a multi-step process that begins with migration of cancer cells from the primary tumor into the circulation to reach lymph nodes and the parenchyma of distant organs (*Massagué and Obenauf, 2016*; *Lambert et al., 2017*). In host organs, disseminated cancer cells interact with a tissue microenvironment that includes organ-specific resident cells, immune cells, perivascular niches, extracellular matrix, cytokines, metabolites, and an oxygen concentration range. This environment eliminates the majority of cancer cells that infiltrate the parenchyma from the circulation, and selects for cells that can adapt, survive as latent entities, and form micrometastases that may eventually grow into clinically manifest metastases. The progression from micro- to macrometastasis is thought to entail a dynamic interaction between disseminated cancer cells and the host microenvironment, which determines an organ-specific pattern of metastatic relapse characteristic of each type of cancer (*Obenauf and Massagué, 2015*; *Celià-Terrassa and Kang, 2018*).

**eLife digest** Cancer cells may not limit themselves to the tissue or organ where they first formed. In some cases, the cells can spread to form tumors in new parts of the body. This process is known as metastasis, and because it is difficult to treat it causes the majority of cancer deaths. To develop new treatments, researchers are trying to learn more about the different steps involved in metastasis.

As cancer cells travel through the body they must adapt to the changing environments they encounter, and avoid detection and destruction by the immune system. To do so, they turn different genes on or off. When the cells reach their final destination tissue, they divide to form microscopic clusters, or 'micrometastases', that can grow into new tumors. Micrometastases can sometimes be eliminated by chemotherapy or radiation. Examining which genes are active in the micrometastases may help researchers to find other ways to kill these cancer cells before they can grow into larger tumors that are harder to treat.

Basnet et al. have developed a new tool called Flura-seq that documents which genes are active in small clusters of cells in the tissues of living animals. The tool was used to study how breast cancer cells form new tumors in the lungs and brains of mice. The results of the study reveal that lung and brain micrometastases have different patterns of gene activity. In particular, the cancer cells in the lungs turn on antioxidant genes. If they did not, they were killed by a condition known as oxidative stress. This suggests that hindering the activity of the antioxidant genes could help to stop tumors forming in the lungs.

Further studies that use the new Flura-seq technique could help researchers to learn more about the early stages of cancer and cancer metastasis. The technique could also be used to study gene activity in other small groups of cells as tissues develop and regenerate.
DOI: https://doi.org/10.7554/eLife.43627.002

Overt metastasis is associated with high morbidity and mortality, and is a major clinical concern. Large metastatic lesions accumulate genetic and epigenetic alterations and stably express specific transcriptional signatures (*Easwaran et al., 2014*; *Roe et al., 2017*). In recent years, analysis of these signatures in cells derived from human tumors and xenografts has uncovered numerous factors whose expression mediates organ-specific metastasis in animal models and is associated with organ-specific metastasis in patients (*Ell and Kang, 2013*; *Kang et al., 2003*; *Minn et al., 2005*; *Bos et al., 2009*; *Boire et al., 2017*; *Tavazoie et al., 2008*; *Valiente et al., 2014*; *Chen et al., 2016*; *Shibue et al., 2012*; *Bragado et al., 2013*; *Gao et al., 2016*). Some of these mediators serve as targets of therapeutic intervention against metastatic cancer (*Celià-Terrassa and Kang, 2018*; *Sleeman and Steeg, 2010*). By contrast, cancer cells in the early stages of metastatic colonization may dynamically alter their gene expression profiles in response to specific stresses experienced in distant organs as they adapt to the host tissue microenvironment and form long-lasting metastatic seeds. These early disseminated cells represent a crucial transition state and may be particularly vulnerable to therapy since they can sometimes be eliminated using adjuvant therapy after surgical resection of primary tumors, unlike established macrometastases. Thus, it is critical to understand the vulnerabilities, dynamic as they may be, of early micrometastases. However, insight into the dynamic early micrometastatic state has been limited by the lack of sensitive techniques for in situ transcriptomic analysis of minute numbers of disseminated cancer cells within large host organs.

Current techniques to study cell-type-specific transcriptomes have limitations that preclude their effective application in studying metastasis-initiating cancer cell populations. Single-cell RNA sequencing (scRNA-seq), with or without an intervening fluorescence activated cell sorting (FACS) step, allows identification of the transcriptomes of underrepresented cell populations at a single-cell level, but it requires extensive physical and enzymatic processing of the tissue, which disrupts the effects of the host microenvironment while exerting stress on these cells, thus compromising the ability to discern the impact of the host stroma from the transcriptome of the isolated cells. Furthermore, only about 10–20% of the transcripts are captured during the library preparation in scRNA-seq which severely limits the coverage of transcriptome of cells of interest (*Hwang et al., 2018*). In addition, scRNA-seq is challenging to apply in tissues and cell types that are difficult to dissociate

into single cells. In situ transcriptomic profiling obviate these problems but lack the necessary sensitivity for disseminated cancer cells that represent less than 1% of the tissue cell population. For example, translating ribosome affinity purification and mRNA sequencing (TRAP-Seq) (*Heiman et al., 2008*) is not suitable to analyze cells that constitute less than 1% of the total population (*Bertin et al., 2015*; *Obenauf et al., 2015*). Direct-enzyme-based metabolic tagging of RNA with thiouracil (TU) and ethynyl cytosine (EC) in the cells of interest are limited in sensitivity and specificity due to collateral tagging and purification of tagged RNA in cells lacking the enzymes, and requires additional in vitro biotinylation steps (*Cleary et al., 2005*; *Gay et al., 2014*; *Gay et al., 2013*; *Miller et al., 2009*; *Hida et al., 2017*). TU tagging has a sensitivity limit of 5% (*Gay et al., 2013*). Thiol (SH)-linked alkylation of the metabolic labeling of RNA in tissue (SLAM-ITseq) eliminates the noise associated with the purification of RNAs that are not thiol tagged in TU-tagging method (*Matsushima et al., 2018*), but undesired TU tagging through endogenous enzymes in cells lacking UPRT expression remains a limitation. Other methods such as laser capture microdissection/RNA-seq are useful in preserving the spatial information (*Nichterwitz et al., 2016*), however, require sophisticated tools and are challenging to use in rare cell populations that are sparsely distributed in the tissue.

Here, we describe the development of a CD-based method for in situ transcriptomic profiling of rare cell populations with high sensitivity (less than 0.01% of an organ), and the application of this method to the analysis of organ-specific micrometastatic adaptation. Using this approach, we define microenvironment-dependent transcriptional programs in micrometastatic pulmonary and brain metastases from breast cancer, identify oxidative stress as a lung-specific liability of disseminated cancer cells, and demonstrate that NRF2 activation and upregulation of distinct antioxidant genes are adaptive responses to this stress in lung micrometastases. This oxidative stress and adaptive transcriptional events are reversible upon removal of metastatic cells from the tissue microenvironment, and disappear when metastasis-derived cells are placed in culture. We validate our findings in metastatic tumors from different organ sites from patients with breast cancer. Thus, Flura-seq identifies both a dynamically induced organ-specific stress program activated by metastasis-initiating cancer cells in the pulmonary microenvironment, as well as an adaptive transcriptional program that ensures cancer cell survival, which could be targeted to therapeutic advantage.

## Results

### 5-FU tagging allows isolation and quantitation of variable abundance transcripts

Cytosine deaminase (CD) is a key enzyme of the pyrimidine salvage pathway in fungi and prokaryotes, but is absent in mammalian cells, which instead use cytidine deaminase for the same purpose (*Mullen et al., 1992*). In addition to converting cytosine to uracil, CD can also convert 5-fluorocytosine (5-FC), a non-natural pyrimidine, to 5-fluorouracil (5-FU). 5-FU is endogenously converted to fluorouridine triphosphate (F-UTP), which is incorporated into RNA (*Figure 1A,B*). An antibody-based purification step that specifically captures the 5-FU-tagged RNA would yield a sample suitable for sequencing. Although 5-FU or the combination of CD expression and 5-FC treatment are cytotoxic in some cells (*Austin and Huber, 1993*; *Kievit et al., 1999*; *Longley et al., 2003*), such toxicity requires more than 7 days of treatment (*Hamstra et al., 2004*; *Kaliberov et al., 2006*). We hypothesized that short-term 5-FC treatment in CD-expressing cells may avert such toxic effects and minimize transcriptional distortion, thus allowing in situ transcriptomic profiling of rare cell populations.

We expressed *S. cerevisiae* CD in human embryonic kidney 293 T cells (293 T-CD cells), and treated the cells with 5-FC to yield intracellular 5-FU, which is incorporated into newly synthesized RNA. Antibodies against bromodeoxyuridine (BrdU) crossreact with other halogenated uridines incorporated into nucleic acids (*Aten et al., 1992*). Accordingly, untransfected control cells incubated with 5-FU showed positive anti-BrdU immunofluorescence, whereas cells incubated with 5-FC did not (*Figure 1—figure supplement 1A*). The anti-BrdU antibody also stained 293 T-CD cells when treated with 5-FC, demonstrating that the antibody binds to exogenous or CD-generated 5-FU derivatives but not 5-FC derivatives (*Figure 1—figure supplement 1A*). To test the specificity and efficiency of RNA isolation, we immunoprecipitated messenger RNA (mRNA) from 5-FU-labeled cells with the anti-BrdU antibody and determined the mRNA levels of representative high expression

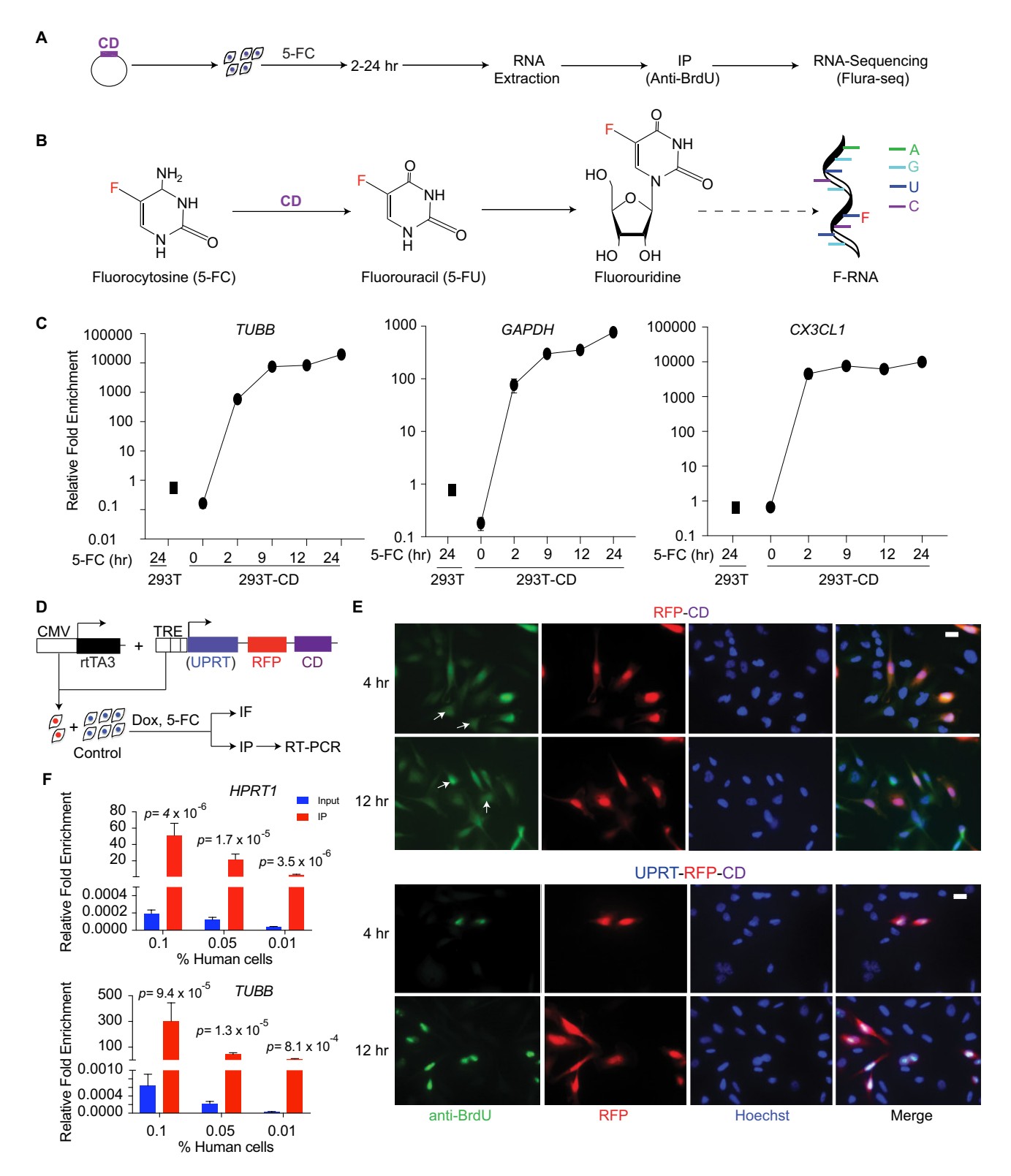

**Figure 1.** Cell-type-specific labeling and isolation of RNAs by Flura-tagging. (**A**) Schematic diagram showing RNA labeling and isolation using CD and 5-FC; (**B**) Chemical reactions steps involved in the labeling of RNA using CD and 5-FC; (**C**) Enrichment of mRNAs immunopurified by anti-BrdU antibody in cells expressing CD relative to WT cells and normalized to their corresponding inputs after 5-FC treatment for the indicated times, as measured by qRT-PCR for the representative genes ($n = 3,\pm$S.E.); (**D**) Schematic diagram of the constructs used for inducible expression of UPRT and/or CD, and the

*Figure 1 continued on next page*

*Figure 1 continued*

experimental design of Flura-tagging; (**E**) MDA231 cells expressing RFP-IRES-CD or UPRT-T2A-RFP-IRES-CD were co-cultured with unmodified control cells, treated with 5-FC, and Flura-tagging was assessed by BrdU immunostaining (*n* = 3, Scale bar, 20 µM). Arrow indicates cells lacking CD expression but stained with BrdU antibody; (**F**) 100, 500 or 1000 human MDA231 cells expressing CD/UPRT were co-cultured with $10^6$ mouse 4T1 cells, treated with 5-FC for 12 hr, and 5-FU-tagged RNAs were immunoprecipitated. The fold enrichment of the indicated representative human genes over mouse housekeeping genes (*mHPRT1*) was measured by qRT-PCR (*n* = 3 ± S.E.). p-Values were calculated by unpaired two-tailed student's t test.

DOI: https://doi.org/10.7554/eLife.43627.003

The following figure supplement is available for figure 1:

**Figure supplement 1.** Cell-type-specific labeling and isolation of RNAs by cytosine-deaminase-based 5-FU tagging.

DOI: https://doi.org/10.7554/eLife.43627.004

genes (*glyceraldehyde 3-phosphate dehydrogenase, GAPDH; tubulin beta chain, TUBB*) and low expression genes (*chemokine CX3C motif ligand 1, CXC3L1*) by reverse transcriptase-polymerase chain reaction (RT-PCR). In 293 T-CD cells, these mRNAs were detectable after 2 hr of treatment with 5-FC, and the levels continued to increase for up to 24 hr (*Figure 1C*). The relative enrichment of the RNAs was two to three orders of magnitude higher in 293 T-CD cells compared to control 293 T cells (*Figure 1C*). These results demonstrate that 5-FU tagging allows specific labeling and purification of newly synthesized transcripts.

## Cell specificity of RNA Flura-tagging

5-FU can be transported across cell membranes based on its concentration gradient (*Ojugo et al., 1998*; *Wohlhueter et al., 1980*). Therefore, we determined whether 5-FU labeling of RNAs using this method would be restricted to CD-expressing cells or collaterally affect neighboring cells. We generated CD-expressing derivatives of MDA-MB-231 (MDA231) cells, a cell line derived from the pleural fluid of a patient with highly metastatic, triple hormone receptor-negative breast cancer (*Cailleau et al., 1974*). The CD-expressing derivative cells, MDA231-CD, were co-cultured with unmodified MDA231 cells (*Figure 1D*), incubated with 5-FC, and the 5-FU labeling of individual cells was determined based on anti-BrdU immunofluorescence. The co-cultures showed 5-FU-labeling not only in MDA231-CD cells but also in unmodified MDA231 cells (*Figure 1E*).

To limit the diffusion of 5-FU from CD-expressing cells, we implemented a dual strategy. First, we engineered MDA231 cells to co-express CD and uracil phosphoribosyl transferase (UPRT). UPRT directly converts 5-FU to 5-fluorouridine monophosphate (F-UMP), which does not diffuse across cell membranes, bypassing the generation of 5-fluorouridine (*Figure 1—figure supplement 1B*). We developed a polycistronic vector that allows doxycycline (Dox)-inducible co-expression of UPRT, CD and red fluorescence protein (RFP) (*Figure 1D*), and transduced this vector into the cells (MDA231-CD/UPRT cells). Second, since thymine can competitively inhibit cellular uptake of 5-FU (*Yuasa et al., 1996*), we included thymine in the medium as a competitive inhibitor of 5-FU transport. This dual strategy restricted the anti-BrdU immunostaining to cells expressing CD (*Figure 1E*). Thymine was used in all subsequent in vitro and in vivo experiments.

Next, we determined whether this 5-FU-tagging method, 'Flura-tagging', could be used to isolate RNA specifically from cells of interest that were admixed with a large proportion of unlabeled cells. MDA231-CD/UPRT cells were co-cultured with 4T1 mouse breast cancer cells at ratios of $10^{-3}$ to $10^{-4}$ (100 to 1000 MDA231-CD/UPRT cells to $10^6$ 4T1 cells). After 12 hr of incubation with 5-FC, 5-FU-labeled mRNAs were immunoprecipitated with anti-BrdU antibody, and the proportion of human and mouse mRNA for representative housekeeping genes was determined by qRT-PCR. Notably, human mRNAs were enriched by more than 10-fold relative to mouse mRNAs, despite human cells comprising 0.01–0.1% of the total cell population (*Figure 1F*). These results demonstrated the efficacy and specificity of the technique in measuring newly synthesized RNAs from small cell populations of interest in a heterogeneous mixture of cells.

To identify potential transcriptional alterations caused by Flura-tagging, we compared the transcriptome of MDA231-CD/UPRT cells treated with two different concentrations of 5-FC (50 µM and 250 µM), with that of untreated cells that do not express CD/UPRT, using global RNA sequencing analysis (RNA-seq). Over 99% of ~20,000 analyzed genes showed statistically similar expression with 50 µM or 250 µM 5-FC (*Supplementary file 1*) indicating that Flura-tagging introduces minimal alteration in the basal transcriptomes of cells in our experimental conditions.

## Flura-tagging system effectively captures signal dependent change in gene expression

To determine whether Flura-tagging could be used to analyze the transcriptional response to extrinsic regulatory signals, we examined the transcriptional response to TGF-β, a pleiotropic cytokine that regulates the expression of many genes involved in diverse cellular processes (*David and Massagué, 2018*). We used the TGF-β response of MDA231 cells (*Padua et al., 2008*) as an indicator of the sensitivity and fidelity of our method. MDA231-CD/UPRT cells were treated with 5-FC and either TGF-β or the TGF-β receptor kinase inhibitor SB-505124 (SB). We subjected total RNA from MDA231 cells and immunoprecipitated 5-FU-tagged RNA from MDA231-CD/UPRT cells to RNA-seq analysis. In MDA231 cells, 176 genes showed either an increase or decrease of more than two-fold in transcript levels upon TGF-β treatment (*Supplementary file 2*). RNA-Seq analysis of Flura-tagged RNA samples ('Flura-seq') captured the TGF-β transcriptional response of MDA231 cells with high accuracy and fidelity, compared to the RNA-seq control (*Figure 2A,B*; *Supplementary file 2*). It is also noteworthy that Flura-seq showed an enhancement in the fold change of the majority of TGF-β induced genes compared to the control (*Figure 2B–D*). This is possibly because Flura-seq only detects newly synthesized transcripts, whereas RNA-seq accounts for the total transcripts and thus dilutes the transcriptional response to an acute TGF-β stimulus. On the other hand, Flura-seq identified 575 genes differentially expressed upon TGF-β treatment (*Supplementary file 2*). Comparison of the genes uniquely identified by Flura-seq (2.5 hr post TGF-β treatment) to the differential gene

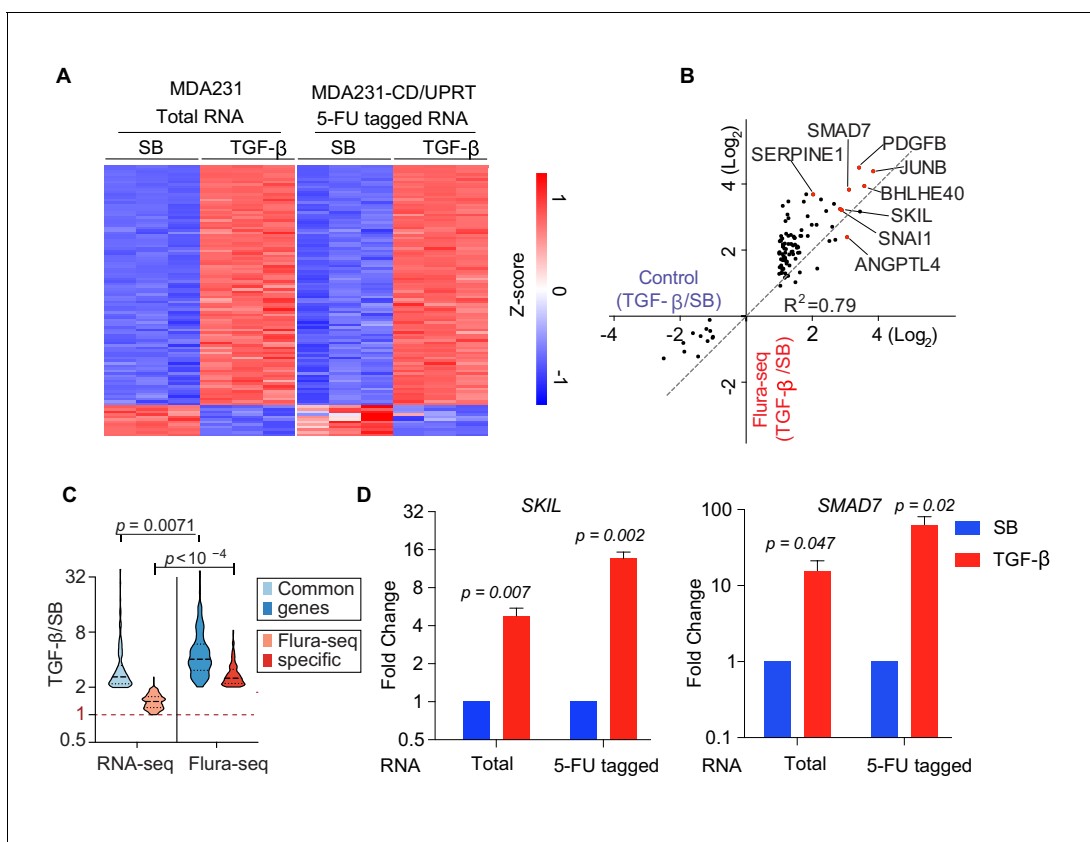

**Figure 2.** Flura-tagging system effectively captures signal dependent change in gene expression. (A–D) MDA231 cells expressing CD/UPRT were treated with 5-FC for 30 min prior to TGF-β or SB-505124 (SB, a TGF-β receptor inhibitor) treatment for 150 min. (A) Change in gene expression in TGF-β-treated cells relative to SB-treated cells as determined by RNA-seq of total RNA from control cells or 5-FU-tagged RNA from Flura-tagged cells. The heat map includes all the genes whose expression changed by more than 2-fold (p<0.01) in response to TGF-β in control cells; (B) Cartesian plot of the data in *Figure 2B*. Each dot represents a gene; typical TGF-β-responsive genes are highlighted (n = 3); (C) Violin plot of the genes induced by TGF-β as identified by RNA-seq and Flura-seq, and Flura-seq only; (D) Expression of the indicated representative TGF-β-induced genes was determined by qRT-PCR in total RNA and in anti-BrdU immunoprecipitate (n = 3,±S.E.). p-Values were calculated by unpaired two-tailed student's t test.
DOI: https://doi.org/10.7554/eLife.43627.005

expression data sets in MDA231 cells 6 hr post TGF-β treatment (*Tufegdzic Vidakovic et al., 2015*) showed that 83 of the genes identified only by Flura-seq were induced by TGF-β as detected by RNA-seq at later time points, suggesting that Flura-seq captures early signal-induced gene expression that is missed by RNA-seq due to dilution by the preexisting basal mRNA pool. Collectively, these results show that Flura-seq can accurately capture global changes in gene expression in response to stimuli.

## Flura-seq analysis of rare metastatic cells in situ

Next, we determined whether Flura-seq could be used to characterize transcriptomics in situ from a small number of cancer cells disseminated in an intact organ that would be challenging to achieve using existing technologies. MDA231 cells expressing a GFP-luciferase fusion protein for imaging and bioluminescence analysis and Dox-inducible CD/UPRT for Flura-seq analysis, were inoculated into the tail vein of $Foxn1^{nu}$ immunodeficient mice to allow colonization of the pulmonary parenchyma (*Figure 3A*). A small proportion of the injected cells survive in the lungs and initiate metastatic outgrowth (*Minn et al., 2005*). At day 31 after inoculation, the cancer cell population was present as micrometastatic colonies throughout the pulmonary parenchyma (*Figure 3—figure supplement 1A,B*). In tissue sections, the size distribution of these colonies ranged from 112 to 877 cells per cluster, with a mean value of 333 cells (*Figure 3—figure supplement 1C*). CD/UPRT expression was induced by doxycycline treatment on day 28, and mice were administered 5-FC (250 mg/kg) and thymine (125 mg/kg) on day 31 for 4 hr to 12 hr before harvesting the lungs for immunoprecipitation of 5-FU-tagged RNAs (*Figure 3A*). The 5-FC dose was selected based on the non-toxic dose of the structurally related thiouracil in mice (250 mg/kg) that has been used for RNA tagging with thiouracil (*Gay et al., 2014*).

We determined the relative fold enrichment of 5-FU tagging in vivo by measuring the relative capture of representative housekeeping human and mouse transcripts. The human mRNAs were enriched more than a 10,000-fold compared to the corresponding mouse mRNAs (*Figure 3B*), indicating that 5-FU tagging occurs primarily in the human cells of interest and that tagged RNAs can be purified efficiently from intact mouse lung tissue. We also compared the relative fold enrichment of 5-FU tagging with TU tagging, an analogous covalent RNA labeling technique (*Gay et al., 2013*; *Miller et al., 2009*). To this end, mice harboring lung micrometastases were treated in parallel with TU for 12 hr according to previous studies (*Miller et al., 2009*). Analysis of tested human mRNAs relative to the mouse mRNAs showed approximately 10-fold enrichment with TU tagging compared to over 10,000-fold enrichment with 5-FU tagging (*Figure 3B*). In parallel, we determined the percentage of human cells present in the mouse lungs in these experiments. Approximately 0.003% to 0.08% of the total cell population comprised of human cells, as determined by RFP expression from the polycistronic UPRT/CD/RFP vector (*Figure 3—figure supplement 1D*). Since one mouse lung contains approximately 150 million cells (*Perrone et al., 2010*), we estimate that RNA from as few as approximately 5000 human cancer cells per mouse lung could be analyzed by 5-FU tagging (*Figure 3—figure supplement 1E*).

To determine whether 5-FU tagged mRNA from micrometastatic lesions could be used to characterize the in situ transcriptome of cancer cells, mice were treated with 5-FC for 4 hr or 12 hr, and tagged RNAs were immunopurified and sequenced. The sequenced reads were aligned to a hybrid genome containing both human and mouse genomes, so that reads coming from human or mouse cells could be distinctly identified. In mice treated with 5-FC for 4 hr, approximately 53% of the aligned reads were mapped to human genome, whereas 74% of the aligned reads were mapped to human genome when the mice were treated with 5-FC for 12 hr (*Figure 3C*). Fewer than 1% of the mapped reads in the non-immunopurified input samples were aligned to the human genome while 99% of the reads aligned to the mouse genome (*Figure 3C*).

To further distinguish transcripts derived from the cells of interest (human cells) versus other cells (mouse cells), we focused on transcripts that were enriched more than 2-fold relative to input. After applying this enrichment cut-off, the reads were aligned to 7487 human genes and 231 mouse genes (*Figure 3D*). When the cutoffs were increased to 4, 8 and 16-fold, the number of human genes identified remained the same, whereas the mouse genes were completely eliminated (*Figure 3D*). These results demonstrate the sensitivity and specificity of Flura-seq in identifying in situ transcriptomes of cells of interest in vivo (*Figure 3E*).

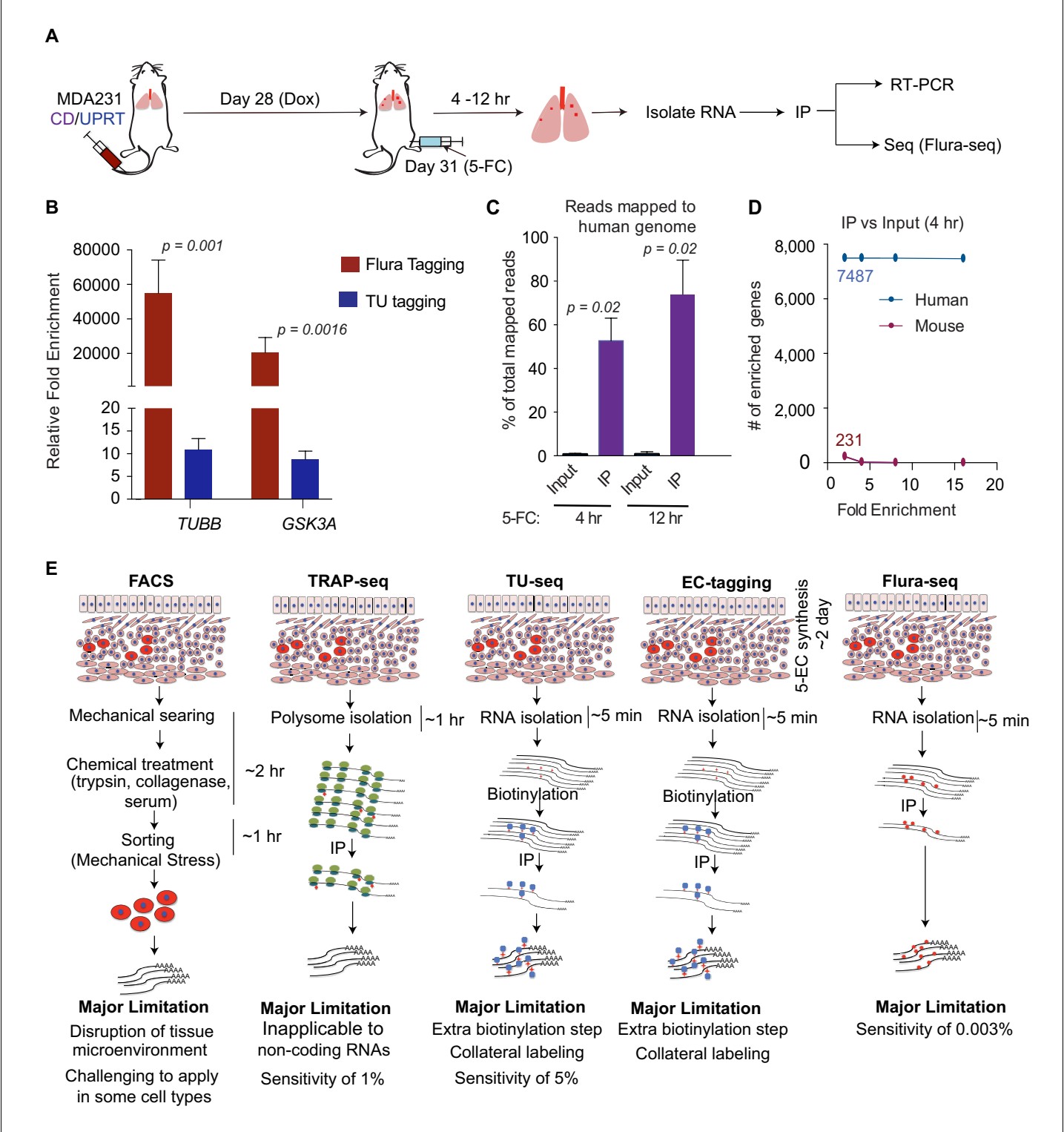

**Figure 3.** Flura-tagging of rare metastatic cells in situ. (**A**) Schematic diagram of lung colonization xenograft assay used for evaluation of Flura-tagging in vivo. Athymic mice were injected through the tail vein with 50,000 MDA231 cells expressing CD/UPRT and GFP-luciferase. After 4 weeks, mice were treated with doxycycline (3 days) to induce CD/UPRT expression in the disseminated cancer cells, and injected with 5-FC. Lungs were harvested 4 hr to 12 hr later, and subjected to immunopurification of 5-FU-tagged RNA for RNA-seq analysis (Flura-seq); (**B**) Comparison of relative fold enrichment of Flura-tagging and TU-tagging in vivo after immunoprecipitation. Mice with CD/UPRT expressing MDA231 lung metastases were injected with either 5-FC or TU for 12 hr, lungs were harvested. Flura-tagged RNA was purified by immunoprecipitation, and TU-tagged RNA was biotinylated and purified by

*Figure 3 continued on next page*

*Figure 3 continued*

streptavidin beads. The relative fold enrichment of representative human housekeeping genes relative to representative murine housekeeping genes (*mHPRT1*, *mLDH1*, *mPGK1* and *mGAPDH*), normalized to their corresponding inputs, were determined by qRT-PCR (*n* = 5,±S.E.); (**C**) Flura-seq specifically enriches for 5-FU-tagged human transcripts from lung micrometastases. 5-FU-tagged RNA from mouse lungs bearing CD/UPRT-expressing MDA231 cells and treated with 5-FC for 4 hr or 12 hr were immunopurified and sequenced. RNA reads were aligned to a hybrid genome containing the human and mouse genomes. The percentage of aligned reads mapped to human genome for the Flura-seq samples and the corresponding unprecipitated input is shown (*n* = 2,±S.E.); (**D**) Number of human and mouse genes identified by Flura-seq (samples with 4 hr of 5-FC treatment) at different fold enrichment cutoffs relative to the corresponding unprecipitated inputs (*n* = 2); (**E**) Comparison of the workflow, limitations and sensitivity of Flura-seq versus other methods for transcriptomic analysis of rare cell populations in tissues. p-Values were calculated by unpaired two-tailed student's t test.

DOI: https://doi.org/10.7554/eLife.43627.006

The following figure supplement is available for figure 3:

**Figure supplement 1.** Flura-tagging of rare metastatic cells in situ.

DOI: https://doi.org/10.7554/eLife.43627.007

## Flura-seq identifies organ-specific in situ transcriptomes in micrometastases

Next, we applied Flura-seq to define the in situ transcriptomes of breast cancer cells during early stages of metastatic colonization in distinct microenvironments of the brain and lungs. MDA231-CD/UPRT cells were injected intracardially into the arterial circulation of female mice to allow infiltration of multiple organs (*Figure 4A*). In the lungs and brain, the cells developed micrometastases within 31 days of injection (*Figure 4—figure supplement 1A*). The cancer cells were also injected into the mammary fat pad (MFP) to generate orthotopic mammary tumors (*Figure 4A*). To identify the genes that are expressed in response to the organ-specific microenvironment, we harvested the brain, lungs, and mammary tumors, and subjected samples to Flura-seq analysis. In parallel, an aliquot of these tissue samples was dissociated into single cells and cultured in selective media to isolate the labeled MDA231 cells as previously described (*Minn et al., 2005*). Following selection and in vitro expansion for 1–2 weeks (passage 2), these cultures were subjected to RNA-seq analysis (*Figure 4A*). Principal component analysis (PCA) revealed that the in situ transcriptomes of MDA231 cells in different tissues were highly divergent from one other (*Figure 4B*). In contrast, in vitro culture of the mammary tumor and metastasis-derived cells diminished their transcriptomic differences (*Figure 4B*).

Flura-seq identified several thousand genes that were differentially expressed in different tissues whereas the same cells showed differential expression of only a few hundred genes when cultured in vitro (*Figure 4C*, *Supplementary file 3*). The majority of organ-specific gene expression changes were not preserved when the cells were isolated from the host tissues and expanded in culture. These results suggested that micrometastases have considerable transcriptional plasticity and dynamically regulate gene expression in response to microenvironmental cues. In situ transcriptomic analysis is therefore critical to capture the phenotypic state of micrometastatic cells in the biologically relevant intact tissue context.

## Mitochondrial complex I expression and oxidative stress in lung micrometastatic cells

Analysis of in situ organ-specific transcriptomes unexpectedly revealed that lung micrometastases had the highest content of unique transcriptional activity relative to brain micrometastases and mammary tumors, suggesting that distinct requirements exist for successful metastasis initiation in the lung microenvironment (*Figure 4D*, *Figure 5—figure supplement 1A*). Gene Ontology (GO) analysis of the differentially expressed cancer cell genes in the different tissues revealed that genes encoding components of the mitochondrial electron transport chain, particularly genes encoding Complex I subunits, were significantly upregulated in lung metastases relative to both brain metastases and orthotopic mammary tumors (*Figure 5A*). Gene set enrichment analysis (GSEA) further confirmed the upregulation of Complex I-encoding genes in lung micrometastases (*Figure 5B*). The enrichment of these genes was not observed when the cancer cells were isolated from each organ and cultured in vitro under similar conditions (*Figure 5—figure supplement 1B*), suggesting that the lung microenvironment drives Complex I expression in metastatic cells. In fact, Complex I genes were

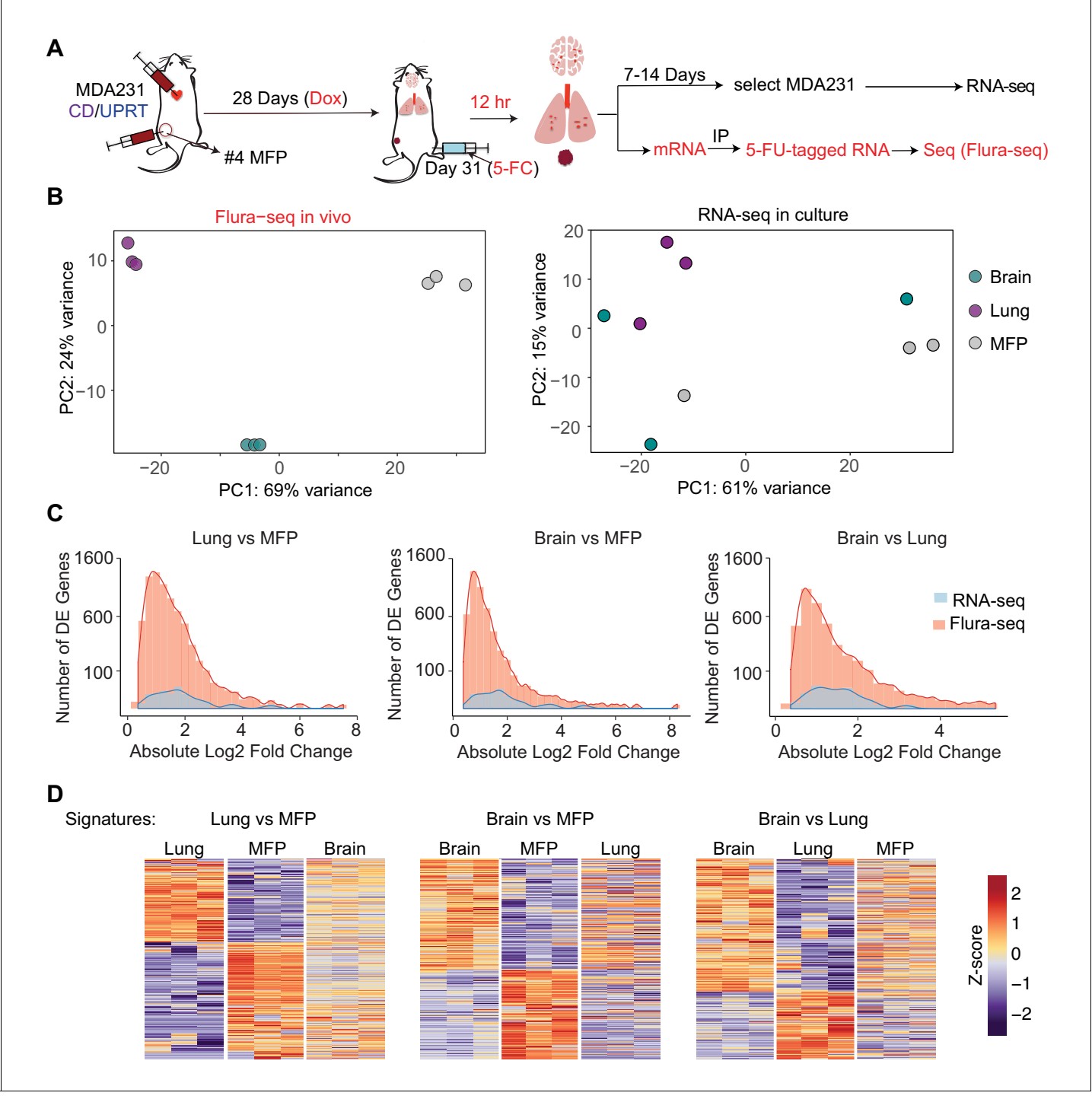

**Figure 4.** Flura-seq identifies organ specific in situ transcriptomes in micrometastases. (**A**) Schematic diagram of experimental design used to obtain tissue specific transcriptomes of MDA231 cells in mice; (**B**) Principal component analysis of genes expressed by MDA231 cells in the indicated organs, as determined by Flura-seq of fresh tissue, or by RNA-seq of in vitro cultured cells derived from these tissues; (**C**) Comparison of differentially expressed genes in metastatic MDA231 cells in different organs as determined by Flura-seq of fresh tissue versus RNA-seq of tissue-derived MDA231 cell cultures. The number of differentially expressed genes and their corresponding fold-change in the indicated organ pairs were plotted for both methods; (**D**) Heatmap representation of differentially expressed genes identified by Flura-seq in MDA231 cells residing in the indicated pairs of organs, compared to the expression of these genes in the third organ.

DOI: https://doi.org/10.7554/eLife.43627.008

The following figure supplement is available for figure 4:

*Figure 4 continued on next page*

*Figure 4 continued*

**Figure supplement 1.** Flura-seq identifies organ-specific in situ transcriptomes in micrometastases.
DOI: https://doi.org/10.7554/eLife.43627.009

underexpressed in lung metastasis-derived cells in culture relative to cells derived from brain metastases or mammary tumors, possibly due to re-adaptation of the cells when removed from the lung microenvironment.

Complex I activity is a source of reactive oxygen species (ROS) (*Balaban et al., 2005*; *Murphy, 2009*), which at high concentrations cause oxidative stress owing to chemical alteration of proteins and nucleic acids in the cell (*Liou and Storz, 2010*; *Liou and Storz, 2015*). 4-Hydroxynonenal (4-HNE), a product derived from lipid peroxidation in cells, is a marker of oxidative stress (*Liou and Storz, 2015*). A higher level of 4-HNE was present in lung micrometastases compared to the brain micrometastases, as determined by anti-4-HNE immunohistochemistry (*Figure 5C*), indicating higher oxidative stress in the lung micrometastases. Cells counteract the cytotoxic effect of oxidative stress by upregulating genes that have antioxidant activity (*Espinosa-Diez et al., 2015*). Indeed, analysis of the expression of 63 genes that include all the antioxidant enzymes and the proteins that directly detoxify ROS (*Gelain, 2009*) revealed that a set of antioxidant genes were specifically upregulated in the lung micrometastases (*Figure 5D*). To confirm that the transcriptional changes identified reflect changes in protein levels, we performed immunohistochemistry for one of these gene products, glutathione peroxidase 1 (GPX1), which functions in the detoxification of hydrogen peroxide. Anti-GPX1 immunohistochemistry analysis confirmed high expression GXP1 in lung micrometastases compared to brain micrometastases (*Figure 5E*). We also tested whether the organ-specific oxidative stress and antioxidant programs are specific to triple negative breast cancer by analyzing lung and brain micrometastases formed by HCC1954 cell line, a HER2$^+$ human breast cancer cell line. The higher oxidative stress and increased expression of antioxidants were also detected in lung micrometastases relative to brain micrometastases in HCC1954 xenograft model (*Figure 5—figure supplement 2A–C*), indicating that higher oxidative stress and elevated antioxidant program are more general phenomena of early stage lung metastasis in breast cancer.

During oxidative stress, the transcription factor nuclear factor erythroid 2-related factor 2 (NRF2) is stabilized, enabling transcription of an antioxidant transcriptional program (*Ma, 2013*). Lung micrometastases contained high levels of NRF2 compared to brain micrometastases, based on anti-NRF2 immunohistochemistry (*Figure 5F*). To determine whether NRF2 transcriptional activity is increased in lung micrometastases, we created a list of 24 NRF2 target genes based on NRF2 chromatin immunoprecipitation-sequencing data curated by Cistrome database (*ENCODE Project Consortium, 2012*) (*Supplementary File 4*), and performed GSEA analysis on our cancer cell transcriptomes. Indeed, the NRF2 signature was enriched in lung micrometastases compared to brain micrometastases and mammary tumors (*Figure 5G*). Like the Complex I genes, the NRF2 responsive genes were underexpressed in lung metastasis-derived cells placed in culture (*Figure 5—figure supplement 1C*). Collectively, these results show a specific upregulation of Complex I associated with oxidative stress and a strong NRF2 response in breast cancer cells that survive as lung micrometastases.

## Organ-specific oxidative stress in human breast cancer lung metastases

We investigated Complex I gene expression, and the associated oxidative stress and antioxidant responses in breast cancer patients with metastasis. We analyzed RNA-seq data from breast primary tumors and matched lung metastases from 11 patients (*Siegel et al., 2018*). The lung metastases showed significantly higher expression of Complex I genes compared to mammary tumors (*Figure 6A*). Matched pair comparison showed that 73% (8/11) patients had higher expression of Complex I genes in lung metastases than in their matched primary tumors (*Figure 6B*). 100% (8/8) of the patients with higher Complex I genes had higher expression of lung-specific antioxidant genes identified by Flura-seq (*Figure 6B*), and 88% (7/8) of the patients had higher NRF2 gene signature expression (*Figure 6B*).

A closer examination of differentially expressed genes (>2 fold) in lung metastases compared to their corresponding primary tumors revealed that 45% (5/11) patients overexpressed 26–39 out of

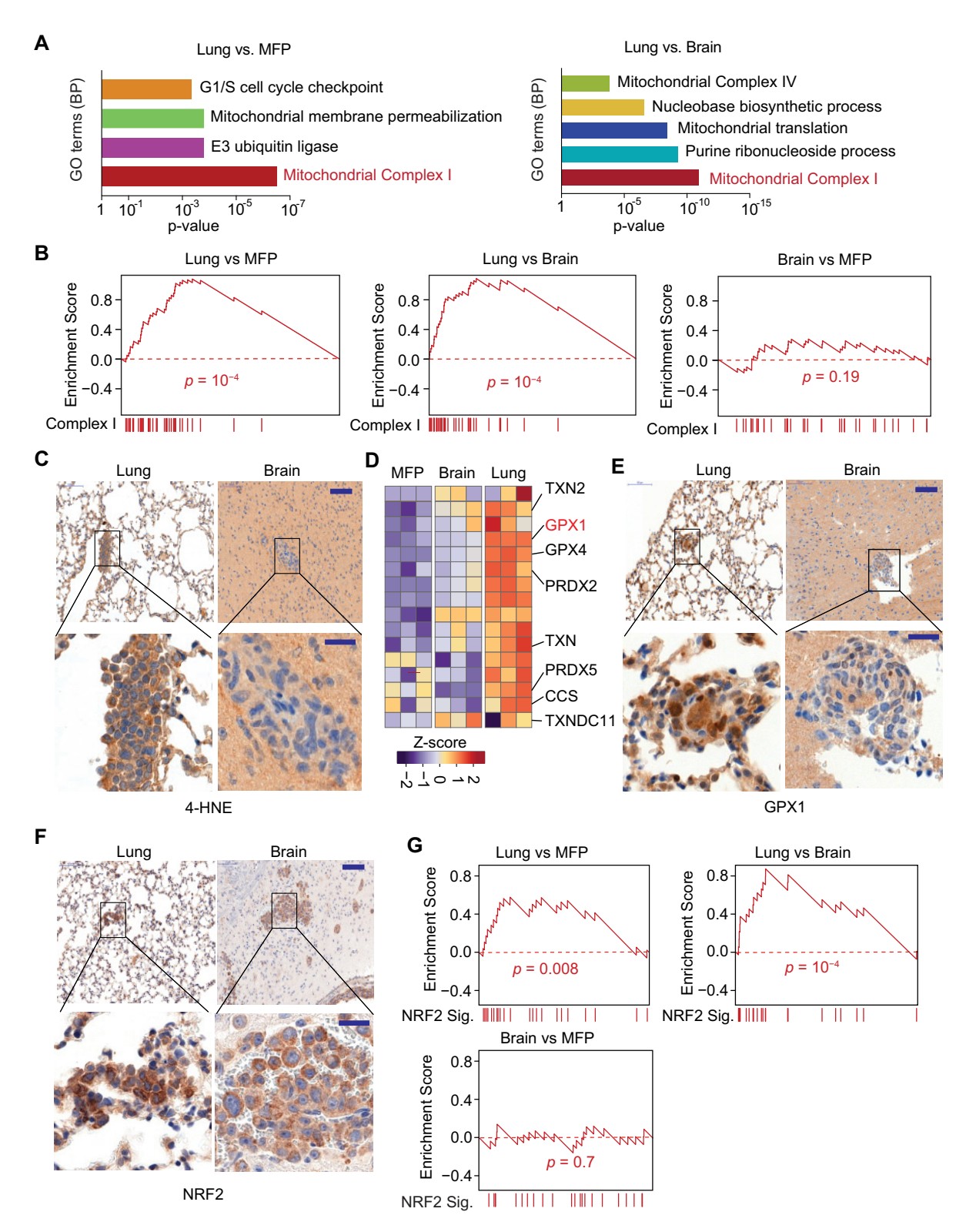

**Figure 5.** Mitochondrial Complex I expression and oxidative stress in lung micrometastases. (**A**) Gene Ontology (GO) analysis of biological processes (BP) of genes that were upregulated in MDA231 lung micrometastases compared to brain micrometastases or mammary tumors. The top functional groups and their corresponding pvalues are shown ($n$ = 3); (**B**) Gene Set Enrichment Analysis (GSEA) analysis of nuclear Complex I genes was performed for the genes identified by Flura-seq in the indicated pairs of MDA231 lung and brain micrometastases and mammary tumors. p-Values were

*Figure 5 continued on next page*

*Figure 5 continued*

calculated by random permutations; (C) Oxidative stress in lung and brain tissue sections containing micrometastases were examined by IHC using anti-4-HNE antibody. Scale bars, 100 µm (top) and 20 µm (bottom); (D) Heatmap representation of the expression of genes encoding known antioxidant factors in MDA231 tumors from the indicated organs. The highlighted genes were also upregulated in clinical samples of lung metastasis from breast cancer patients (*Figure 6D*); (E) IHC analysis of GPX1, an antioxidant gene product identified by Flura-seq to be selectively upregulated in lung micrometastases. Scale bars, 100 µm (top) and 20 µm (bottom); (F) IHC analysis of NRF2 in lung and brain micrometastases. Scale bars, 100 µm (top) and 20 µm (bottom); (G) GSEA analysis of the NRF2 response gene signature applied to Flura-seq data from the indicated pairs of MDA231 lung and brain micrometastases and mammary tumors (*n* = 3). p-Values were calculated by random permutations.

DOI: https://doi.org/10.7554/eLife.43627.010

The following figure supplements are available for figure 5:

**Figure supplement 1.** Differential gene expression in brain and lung micrometastatic cells.

DOI: https://doi.org/10.7554/eLife.43627.011

**Figure supplement 2.** Oxidative stress and antioxidant programs are elevated in lung micrometastases relative to brain micrometastases in HCC1954 xenograft metastasis model.

DOI: https://doi.org/10.7554/eLife.43627.012

43 nuclear encoded Complex I genes (*Supplementary File 5*). We divided these patients into two groups: a high Complex I group of five patients with upregulation of more than 25 Complex I genes, and a low Complex I group of remaining six patients. Complex I high patients were specifically associated with higher expression of lung antioxidant genes and NRF2 signature genes (*Figure 6C,D*), supporting the conclusion that the high expression of Complex I in lung metastasis is associated with the expression of compensatory antioxidant programs. Moreover, eight antioxidant genes that were upregulated together with Complex I genes in patients' lung metastases (*Figure 6D*) were also upregulated in Flura-seq transcriptomes from experimental lung micrometastases (refer to *Figure 5D*).

Finally, we sought to determine whether the differences in oxidative stress and antioxidant responses in lung vs. brain metastases were conserved in clinical samples from breast cancer patients. We performed immunohistochemistry for 4-HNE and NRF2 on a tissue microarray (TMA) containing lung metastases and brain metastases from more than 40 breast cancer patients. Consistent with the Flura-seq findings, 93% (42/45) of the lung metastases scored high for 4-HNE immunostaining, whereas only 16% (9/55) of brain metastases did (*Figure 6E*). Likewise, 78% (32/41) of the lung metastases scored high for NRF2 immunostaining versus only 30% (14/48) in the brain metastases (*Figure 6F*). There was a strong association between oxidative stress (4-HNE) and NRF2 protein level in majority of the patients (*Figure 6G*). Collectively, these results demonstrate higher oxidative stress and elevated protective antioxidant program in lung metastases compared to brain metastases in breast cancer patients.

To test if the NRF2 signature genes overexpression in breast cancer tumors correlate with organ-specific metastasis prognosis outcomes, we calculated the Hazard ratio for NRF2 signature genes for lung, brain and bone metastasis in breast cancer patients. We found that the Hazard ratio was significantly different for lung metastasis but not for brain and bone metastasis (*Figure 6H*), indicating that NRF2 overexpression is advantageous for the survival of breast cancer cells in the lungs compared to brain or bone.

## Discussion

### Organ-specific metabolic adaptation of metastasis-initiating cells

Previous studies have identified stable, organ-specific transcriptomic programs in cancer cells that were selected on the basis of their ability to form macrometastases and then isolated from these lesions by FACS or in vitro culture prior to transcriptomic analysis (*Roe et al., 2017*; *Kang et al., 2003*; *Minn et al., 2005*; *Bos et al., 2009*; *Boire et al., 2017*; *Chen and Massagué, 2012*; *Malladi et al., 2016*; *Bruns et al., 1999*; *Ikeda et al., 1990*; *Ambrogio et al., 2014*). Although these methods successfully identify heritable transcriptional alterations of clinical relevance, these approaches overlook the dynamic transcriptional states that are dependent on tissue-specific microenvironmental cues. Flura-seq now enables the highly sensitive capture of these dynamic

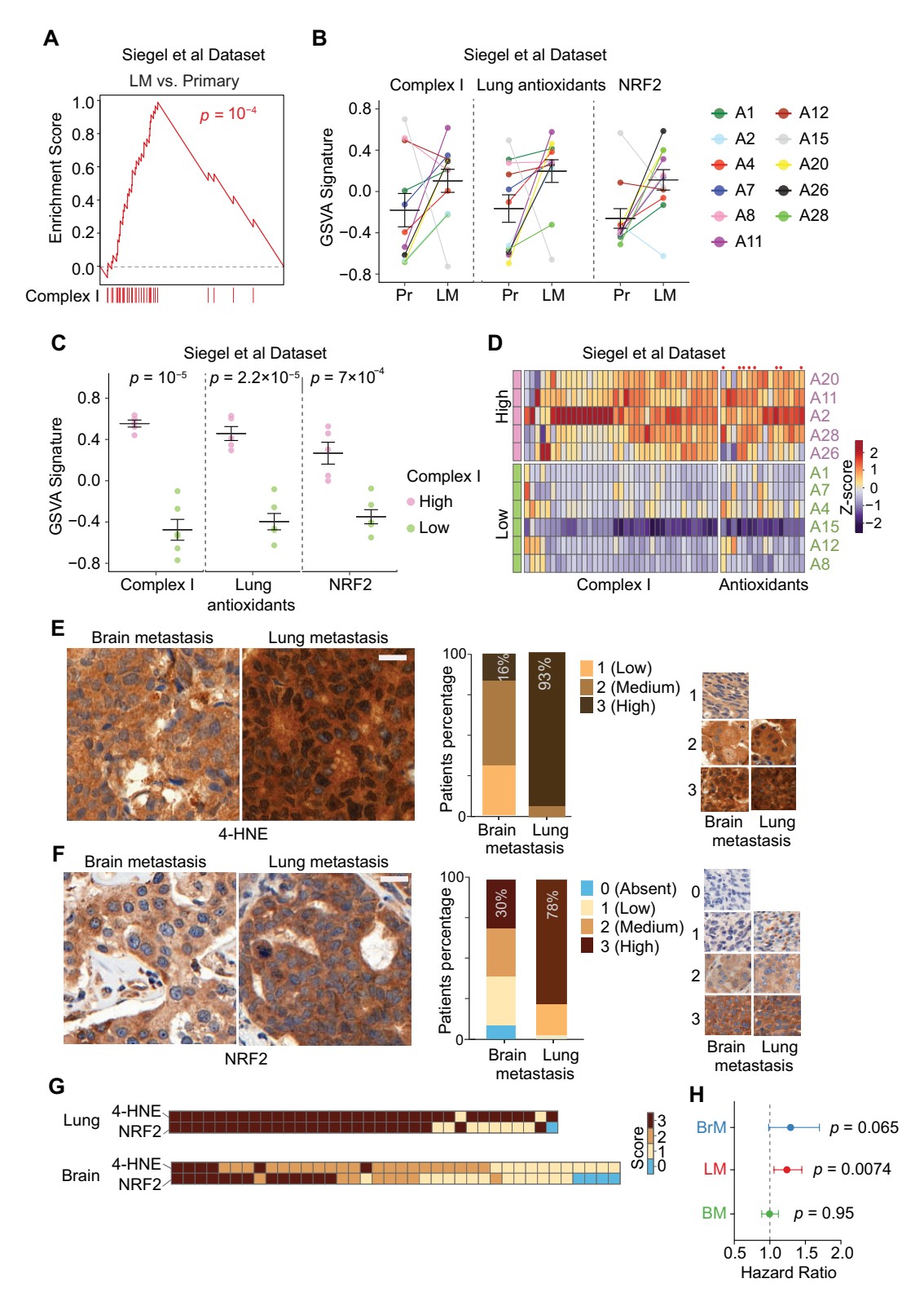

**Figure 6.** Specific oxidative stress in patient-derived lung metastasis tissues. (**A–D**) Expression of nuclear Complex I and antioxidant genes in a gene expression data set of matched primary tumors and lung metastases from patients with breast cancer (*Siegel et al., 2018*). (**A**) GSEA analysis of the expression of Complex I genes shows higher expression of these genes in lung metastases (*LM*) compared to primary tumors (*Primary*); (**B**) Complex I genes, lung antioxidant genes (from *Figure 5D*), and NRF2 response signature genes are upregulated in lung metastases (*LM*) compared to matched

*Figure 6 continued on next page*

*Figure 6 continued*

primary tumor (*Pr*). Gene set variation analysis (GSVA) analysis for transcriptomic data from primary tumors and matched lung metastases of individual patients (letter and color coded); (**C**) Association of anti-antioxidant gene expression with mitochondrial Complex I expression in lung metastasis. Patients were divided into two groups based on the upregulation of Complex I genes in the lung metastases relative to their corresponding primary tumor. The Complex I-High group consisted of five patients with more than 25 out of 43 Complex I genes upregulated by more than twofold in lung metastases relative to the corresponding primary tumor. The Complex I-Low group consisted of six patients with less than 25 Complex I genes upregulated by twofold in the lung metastases compared to the corresponding primary tumor. GSVA signature analysis of Complex I genes, lung antioxidant genes, and NRF2 signature genes was performed in the Complex I-High and -Low groups. p-Values were calculated by unpaired two-tailed student's t test; (**D**) Heatmap of the relative expression of individual mitochondrial Complex I genes and antioxidant genes in lung metastases relative to the corresponding primary tumor. Complex I-High and –Low patient samples are shown as separate groups, in order to highlight the association of antioxidant gene expression with Complex I gene expression. *Red dots*, antioxidant genes that were also identified to be upregulated in mouse lung micrometastases by Flura-seq (shown in ***Figure 5D***); (**E, F**) IHC analysis of oxidative stress marker 4-HNE (**E**) and NRF2 (**F**) in tissue microarrays of brain metastases (*BrM*) and lung metastases (*LM*) from breast cancer patients. Shown are representative images and the quantifications based on the degree of staining (0, no signal: 3, highest signal). (n = 55 samples for BrM and n = 45 for LM for 4-HNE; n = 48 for BrM and n = 41 for LM for NRF2). Scale bar, 20 µm; (**G**) Association between oxidative stress (4-HNE) and NRF2 scores in lung metastases and brain metastases of breast cancer patients. Heatmap of the IHC staining of 4-HNE (**E**) and NRF2 (**F**) was plotted for each patient sample in the TMAs; (**H**) Hazard Ratio plots of the predictive ability of NRF2 signatures in brain (BrM), lung (LM) and bone (BM) metastasis-free survival outcomes in EMC-MSK dataset (GSE2603, GSE5327, GSE2034 and GSE12276). p-Values were calculated using Log-rank test.

DOI: https://doi.org/10.7554/eLife.43627.013

transcriptional states, thus shedding light on crucial adaptive processes underway in micrometastases that could not previously be identified.

In this study, we applied Flura-seq to identify the in situ transcriptomic programs that are differentially active in cancer cells at early stages of metastatic colonization in the lungs and brain. We identified metabolic gene signatures that were specific to the colonized organ and lost upon removing cancer cells from the tissue microenvironment and placing them in culture. Specifically, we identified mitochondrial Complex I as the top upregulated transcriptional alteration in lung metastases that was dynamic and dependent on an intact tissue microenvironment. Elevated expression of Complex I genes correlated with increased oxidative stress and activation of counteracting antioxidant programs including the upregulation of a distinct set of NRF2-driven antioxidant genes in metastatic cells that seed the lungs. Antioxidant and NRF2 activity were also increased in association with high Complex I expression in lung metastases from breast cancer patients, suggesting a role of these pathways in mitigating the cytotoxic effects of oxidative stress on lung metastatic cells. Lung tissue is exposed to higher concentration of oxygen compared to other organs (*Jagannathan et al., 2016*), and high oxygen concentration can cause oxidative stress (*Halliwell, 2014*). It is therefore possible that higher oxygen concentration in the lung micrometastases drives the observed changes. However, we cannot rule out other lung specific microenvironmental cues such as metabolites, cytokines, physical stress, or immune surveillance as sources of the observed changes.

These results demonstrate that metastatic tumor cells arising from a single source adopt unique transcriptional profiles depending on their site of colonization. Despite increasing appreciation that metastatic outgrowths frequently exhibit altered metabolic gene expression compared to their primary tumor counterparts (*LeBleu et al., 2014*; *Dupuy et al., 2015*; *Chen et al., 2007*), whether these metabolic transitions result from the outgrowth of a selected subpopulation predisposed to thrive in a particular location or from the dynamic adaptation of cancer cells to a changing microenvironment remains an open question. Our results support a model wherein tumor cells dynamically adapt to local conditions and suggest that a major determinant of the metabolism of metastatic cells is the site of colonization. These metabolic rearrangements are likely an early event in the establishment of metastatic seeding and may represent a targetable bottleneck against the growth of metastatic lesions.

## Oxidative stress with clinical implications in metastasis

Oxidative stress has been implicated in metastasis, however, the precise role of the stress in metastasis has remained controversial. On one hand, oxidative stress has been observed in cancer cells soon after detachment from epithelia (*Schafer et al., 2009*), and it persists during circulation (*LeBleu et al., 2014*) and upon colonization of metastatic sites in model systems (*Piskounova et al., 2015*; *Gill et al., 2016*). The lung has been proposed to have pro-oxidant environment due to high

oxygen and toxins exposure (*Schild et al., 2018*), and anti-oxidative mediators such as NRF2 (*Wang et al., 2016*; *DeNicola et al., 2015*; *Menegon et al., 2016*), peroxiredoxin 2 (*Stresing et al., 2013*) and thioredoxin-like 2 (*Qu et al., 2011*) stimulate the progression of lung cancer and lung metastasis. On the other hand, ROS has also been reported to promote metastasis, and antioxidants have been shown to inhibit metastasis (*Ferraro et al., 2006*; *Ishikawa et al., 2008*; *Porporato et al., 2014*). The oxidative state and the role of oxidative stress soon after the metastatic cancer cells seed the distant organs before they form macrometastases remain unknown. Our findings demonstrate high oxidative stress in the lung micrometastases of breast cancer, supporting the idea that antioxidant programs promote the progression of lung metastasis and highlighting a critical role for antioxidant mediators in the transition of micrometastases to overt metastases.

Surprisingly, however, our data suggest that elevated antioxidant defenses are not a universal hallmark of metastatic lesions. We found that breast cancer brain metastases experience a low level of oxidative stress and antioxidative response. Given that metastatic cells can exhibit reversible metabolic alterations (*Piskounova et al., 2015*), these results raise the possibility that tumor cells undergo multiple metabolic transitions in order to adapt to the changing microenvironments encountered during the metastatic cascade. Indeed, recent evidence suggests that cancer cells from disparate origins may converge to adopt metabolic phenotypes in a given organ (*Schild et al., 2018*; *Mashimo et al., 2014*). Techniques such as Flura-seq that enable in situ interrogation of tumor cell phenotypes can reveal to what extent these various metabolic transitions are driven by adaptation to the specific microenvironment versus selection of cancer cells with preexisting traits. Given increasing evidence that cell lineage is a critical determinant of cancer cell metabolism (*Mayers et al., 2016*; *Yuneva et al., 2012*) it will be interesting for future studies to determine whether lineage-specific metabolic predispositions contribute to the metastatic organ tropisms of different tumor types. More broadly, these studies will help to shed light on the precise factors in the tissue microenvironment that contribute to organ-specific metabolic profiles.

## Flura-seq as an in situ transcriptomic technique with broad biological applications

Preservation of the intact tissue microenvironment is critical to accurately elucidate the transcriptional state of a cell in vivo. Flura-seq can define in situ transcriptomes from a very rare cell population representing a small fraction (>0.003%) of an organ. The superior sensitivity of Flura-seq compared to related TU-tagging and EC-tagging may be due to the elimination of a biotinylation step and RNA purification system that distinguishes between cytosine derivatives and uracil derivatives. Flura-seq can be easily applied to any cell type that constitutes a rare subpopulation within the host tissue, such as stem cells and specific subtypes of immune and neuronal cells, in addition to residual cancer cells populations during early stages of metastasis or following the shrinking of a tumor with current therapies.

Another feature of Flura-seq is that it only identifies newly synthesized transcripts, which is an advantage in the study of transcriptional responses to cytokines, metabolites, pharmacologic agents, stress signals, and other factors that act by rapidly changing the transcriptomic state of target cells. Further, since Flura-seq involves covalently labeling RNA, it can complement other techniques such as scRNA-seq to combine in situ transcriptomic analysis with profiling of the dissociated cell population with single-cell resolution. Recent advances in scRNA-seq have significantly expanded the application of this technology to the analysis of underrepresented cell types in tissues; however, the method requires extensive physical and enzymatic processing that destroys the tissue microenvironment, and thus microenvironment-dependent gene expression features cannot be accurately captured by scRNA-seq. The higher coverage and applicability of Flura-seq to any tissues and cell types is the principal benefit of Flura-seq over scRNA-seq.

Flura-seq involves the expression of exogenous enzymes, CD and UPRT, and treatment of cells or mice with 5-FC. These treatments may alter the levels of certain transcripts, and is therefore important to validate findings made by Flura-seq with alternative methods such as immunostaining, as shown here. This limitation notwithstanding, Flura-seq provides a sensitive, robust and economical alternative to existing in situ transcriptomics techniques. Thus, the power of Flura-seq in studying rare cell populations can be harnessed to address challenging questions of high biological and clinical significance.

# Materials and methods

## Key resources table

| Reagent type (species) or resource | Designation | Source or reference | Identifiers | Additional information |
|---|---|---|---|---|
| Antibody | anti-BrdU; BrdU antibody (Rat monoclonal) | Abcam | Cat#ab6326 | (1:200) |
| Antibody | anti-CD31; CD31 antibody (Rat monoclonal) | Dianova | Cat#DIA-310 | (1:100) |
| Antibody | anti-GFP; GFP antibody (Chicken monoclonal) | Aves Labs | Cat#GFP-1020 | (1:500) |
| Antibody | anti-4-Hydroxynonenal; 4-HNE antibodyl (Rabbit polyclonal) | Abcam | Cat#ab46545 | (1:75) |
| Antibody | anti-NRF2; NRF2 antibody (Rabbit polyclonal) | Abcam | Cat#ab137550 | (1:600) |
| Antibody | anti-Glutathione Peroxidase 1; GPX1 antibody (Rabbit polyclonal) | Abcam | Cat#ab22604 | (1:200) |
| Antibody | Goat polyclonal anti-chicken | Thermo Fisher | Cat#A-11039 | (1:1000) |
| Antibody | Goat polyclonal anti-rat | Thermo Fisher | Cat#A-11006 | (1:1000) |
| Antibody | Goat polyclonal anti-mouse | Abcam | Cat#ab150117 | (1:1000) |
| Biological sample (Human) | Human breast cancer lung metastases tissue microarray (TMA) | This paper (Section of lung tissue containing cancer cells was surgically removed from breast cancer patients, preserved in paraflim and a small portion of the preserved tumor was used to make the TMA) | N/A | Tissue microarray Available from Edi Brogi |
| Chemical compound, drug | Doxycycline | Sigmal-Aldrich | Cat#D9891 | |
| Chemical compound, drug | 5-Fluorocytosine; 5-FC | Sigma-Aldrich | Cat#F7129 | |
| Chemical compound, drug | 5-Fluorouracil; 5-FU | Sigma-Aldrich | Cat#F6627 | |
| Chemical compound, drug | SB-505124 | Sigma-Aldrich | Cat#S4696 | |
| Chemical compound, drug | Thymine | Sigma-Aldrich | Cat#T0376 | |
| Chemical compound, drug | 4-Thiouracil; TU | Sigma-Aldrich | Cat#440736 | |
| Other | Oligo (dT)$_{25}$ magnetic beads | New England Biolabs | Cat#S1419S | |
| Other | Protein G Dynabeads | Thermo Fisher Scientific | Cat#10004D | |
| Commercial assay or kit | Tissue digestion C-tube | Miltenyi | Cat#130-096-334 | |

*Continued on next page*

Continued

| Reagent type (species) or resource | Designation | Source or reference | Identifiers | Additional information |
|---|---|---|---|---|
| Commercial assay or kit | Mouse Tumor Dissociation Kit | Miltenyi | Cat#130-096-730 | |
| Commercial assay or kit | TruSeq RNA Library Prep Kit v2 | Illumina | RS-122–2001 | |
| Commercial assay or kit | SMARTer PCR cDNA synthesis kit | Clontech | Cat#634926 | |
| Commercial assay or kit | Nextera XT DNA library Preparation Kit | Illumina | FC-131–1024 | |
| Commercial assay or kit | RNeasy MinElute Cleanup kit | Qiagen | Cat#74204 | |
| Commercial assay or kit | cDNA kit-First Strand Transcriptor | Roche | Cat#043790–12001 | |
| Cell line (Human) | MDA231 | Laboratory of Joan Massague | PMID: 19421193 | Expresses TGL |
| Cell line (Human) | MDA231-CD | This paper (MDA231 cells were transduced with rtTA3 and TRE-CD-IRES-RFP) | N/A | Available from Massague lab |
| Cell line (Human) | MDA231-CD/UPRT | This paper (MDA231 cells were transduced with rtTA3 and TRE-UPRT-T2A-RFP-IRES-CD) | N/A | Available from Massague lab |
| Cell line (Human) | 293T | Laboratory of Joan Massague | N/A | |
| Strain, strain background (*Mus musculus*) | Hsd:Athymic Nude- Foxn1$^{nu}$ | Envigo | Cat#069 | |
| Sequence-based reagents | Oligonucleotides | This paper | N/A | Oligonucleotide sequences are provided in *Supplementary file 6* |
| Recombinant DNA reagent | CMV Tight RFP-IRES-CD | This paper (RFP-IRES-CD was subcloned into CMV Tight EGFP Puro (Addgene: Plasmid #26431) vector by removing EGFP). | N/A | Available from Massague lab |
| Recombinant DNA reagent | CMV Tight UPRT-T2A-RFP-IRES-CD | This paper (UPRT-T2A-RFP-IRES-CD was subcloned into CMV Tight EGFP Puro (Addgene: Plasmid #26431) vector by removing EGFP). | N/A | Available from Massague lab |
| Recombinant DNA reagent | rtTA3 | Addgene | Plasmid #26730 | |
| Software and Algorithms | STAR2.5.2b | PMID: 23104886 | https://github.com/alexdobin/STAR | |
| Software and Algorithms | HTSeq v0.6.1p1 | PMID: 20979621 | https://htseq.readthedocs.io/en/release_0.10.0/ | |
| Software and Algorithms | DESeq2 v3.4 | PMID: 25516281 | https://bioconductor.org/packages/release/bioc/html/DESeq2.html | |
| Software and Algorithms | GSVA v3.4 | PMID: 23323831 | https://bioconductor.org/packages/release/bioc/html/GSVA.html | |

## Cell culture

Human embryonic kidney cells transformed with T-cell antigen (293T) and human breast cancer MDA-MB-231 (MDA231) cells were cultured in DMEM High Glucose medium (Wheaton) supplemented with 10% fetal bovine serum and 2 mM L-glutamine. All cell lines have been regularly tested for mycoplasma contamination, and the identity of the cell lines have been authenticated by STR profiling. For the induction of CD or CD/UPRT, cells were treated with 1 μg/ml doxycycline for 24 hr. For 5-FU tagging, cells were treated with 250 μM 5-FC or 5-FU unless indicated. Where indicated, 125 μM thymine was added together with 5-FC. For the induction of TGF-β target genes, cells were treated with 200 pM TGF-β or 2.5 μM SB-505124 for 150 min. For 5-FU-tagging during TGF-β treatment, cells were treated with 5-FC and thymine for 30 min before adding TGF-β or SB-505124.

## Animal experiments

Mouse experiments were performed following the protocols approved by the MSKCC Institutional Animal Care and Use Committee (IACUC). Five- to six-week-old female mice (*Mus musculus*) Hsd: Athymic-Foxn1$^{nu}$ were used in all the experiments. For lung colonization experiments, 50,000 MDA231 cells suspended in 100 μl PBS were injected into the tail vein. For organ-specific metastasis experiments, 50,000 MDA231 cells or 100,000 HCC1954 cells suspended in 100 μl PBS were injected intracardially. For mammary fat pad injection, 50,000 MDA231 cells in 50 μl PBS were mixed with 50 μl matrigel and the mixture was injected in the fat pad of mammary gland #4. Proliferation of injected cancer cells was quantified using bioluminescence imaging following retro-orbital injection of D-luciferin (Gold Biotechnology). CD/UPRT expression was induced by feeding mice doxycycline diet for 2–3 days. For Flura-tagging, mice were injected with 250 mg/kg (500 μl) 5-FC intraperitoneally together with 125 mg/kg (500 μl) thymine subcutaneously. For thiouracil-tagging, mice were injected intraperitoneally with 250 mg/kg (500 μl) of 4-thiouracil. The mice were euthanized 4–12 hr post injection, lungs and brain were harvested and processed for downstream experiments. For RNA analysis, lungs were dissociated using the PRO 200 grinder from PRO Scientific Inc. in RNA extraction lysis buffer. The lung lysates were either used immediately for mRNA extraction or stored at −80°C for later use.

## Immunofluorescence (IF) and Immunohistochemistry (IHC)

For IF, cells were fixed with 4% paraformaldehyde for 10 min, permeabilized with 0.2% TritonX-100 for 10 min, blocked with 5% BSA for 1 hr at room temperature, prior to incubation with primary antibodies at 4°C overnight, and secondary antibodies incubated for 1 hr at room temperature. Mouse lung and brain were fixed in 4% paraformaldehyde 24–48 hr at 4°C, embedded in paraffin and sectioned at 5 μm. Paraffin-embedded sections or tissue microarrays were rehydrated using Histo-Clear (National Diagnostics) followed by 100-95–70% ethanol and water. Antigen retrieval was performed in a steamer for 30 min in citrate antigen retrieval solution. Tissue sections were blocked with 5% normal goat serum for 1 hr, and incubated with primary antibodies overnight. Secondary antibodies conjugated with fluorophores were used for detection. IHC were performed on BOND RX (Leica Biosystems) using standard Epitope Retrieval Solution 2 (Leica Biosystems) for 30 min followed by primary antibody incubation for 30 min and BOND polymer refine detection kit-DAB. Automated image analysis was performed using the FIJI software package. Human histopathological sections were obtained under a biospecimen protocol approved by the MSK Institutional Review Board. All human pathology analyses were performed under the supervision of an experienced breast pathologist (E.B.).

## Flura-tagged and TU-tagged mRNA extraction

Cells or tissues were lysed in lysis buffer (20 mM Tris-HCl pH 7.5, 500 mM LiCl, 1% LiDS, 1 mM EDTA, 5 mM DTT), and mRNAs were extracted using Oligo (dT)$_{25}$ magnetic beads following the manufacturer's protocol. The isolated mRNAs were immunoprecipitated using anti-BrdU antibody (1–5 μg/sample) conjugated with Protein G Dynabeads by overnight incubation at 4°C. The mRNAs were incubated with the antibody bead complex in 0.8X Binding buffer (0.5X Sodium Chloride-Sodium Phosphate-EDTA (SSPE) with 0.025% Tween 20) at room temperature for 1–2 hr in a rotator. Subsequently, beads were washed twice with Binding buffer, twice with Wash buffer B (1X SSPE with 0.05% Tween 20), once with Wash buffer C (TE with 0.05% Tween 20), and once with TE buffer.

The bound mRNAs were eluted in 200 µl of 100 µg/mL BrdU for 45 min in a shaker at room temperature. The eluted RNAs were purified using the RNeasy MinElute Cleanup kit following the manufacturer's protocol. The RNA was eluted in 100 µl RNAase free water. The Flura-tagged RNA elute were re-precipitated as described above, and eluted in 12.5 µl final volume. The RNA was either reverse-transcribed using cDNA kit-First Strand Transcriptor following the manufacturer's protocol, or used for Flura-Seq. TU-tagged mRNAs were purified as described in *Miller et al. (2009)*.

## Isolation of organ-derived cancer cells

Brain, lung or mammary tumors were cut into small fragments (around 1 mm³) and transferred to a tissue digestion C-tube. The tumor pieces were incubated with mouse Tumor Dissociation Kit and further dissociated mechanically on a gentleMACS Dissociator as per manufacturer's protocol. The digestion reaction was stopped with albumin-rich buffer (RPMI-1640 medium containing 0.5% bovine serum albumin (BSA)). A single-cell suspension was obtained by filtering through a 70 µm cell strainer. The cells were then cultured in DMEM High Glucose media containing 10% FBS, 2 mM L-Glutamine, 200 µg/mL Hygromycin and 8 µg/mL Blasticidin to select MDA231 cells.

## Flow cytometry

Harvested lungs were chopped into small pieces (around 1 mm³), which were then incubated at 37°C in 30 mL digestion buffer (5% Fetal Bovine Serum (FBS) 1 mM L-glutamine 0.35 mg/mL Worthington Type III collagenase, $6.25 \times 10^{-3}$ U/mL dispase, 100 U/mL penicillin, 100 µg/mL streptomycin, 6.25 ng/mL amphotericin B) containing 10 mL trypsin and 30 µl DNAse for 1 hr. The cells were filtered through a 70 µM filter, and were collected by centrifugation. The cell pellets were then resuspended in PBS containing 0.1% FBS and 100 µg/ml DAPI, and analyzed using a BD FACS Aria IIU Flow cytometer. CD or CD/UPRT expressing stable cell lines were treated with 1 µg/mL doxycycline for 24 hr, trypsinized, filtered and sorted for RFP positive cells using a BD LSRFortessa Flow cytometer.

## RNA sequencing

RNA-seq library preparation. Total RNA was purified using Qiagen RNeasy Mini Kit. Quality and quantity of the RNA were checked using an Agilent BioAnalyzer 2000. 10 ng of RNA per sample was used for library construction with Sample Prep Kit v2 according to manufacturer's instructions. Libraries were multiplexed on a Hiseq2500 platform, and more than 25 million raw paired-end reads were generated for each sample.

Flura-seq library preparation. RNA was amplified by SMARTer PCR kit with the number of PCR cycles determined empirically based on the amount of purified 5-FU-tagged RNA. The Nextera XT kit was used to prepare sequencing libraries following the manufacturer's protocol. In our in vivo experiments, 20–24 cycles of PCR were used.

## Statistics and data analysis

In all relevant experiments, mice were randomized prior to different treatments. Comparisons between samples were done in the gene expression analysis, and each group had 2–3 biological replicates that are indicated in the figure legends for each experiment. In the in vitro experiments, biological replicates are samples derived from cells that were plated and processed separately. In the in vivo experiments, the biological replicates represent individual mouse. *N* described in the Figure legends represents independent biological replicates. The technical replicates are originated from the same sample but divided into different groups. Sample size for each experiment was determined empirically.

Reads were quality checked using FastQC v0.11.5 and mapped to a human (hg19) or hybrid human-mouse (hg19-mm10) genome with STAR2.5.2b (*Dobin et al., 2013*) using standard settings for paired reads. Uniquely mapped reads were assigned to annotated genes with HTSeq v0.6.1p1 (*Anders and Huber, 2010*) with default settings. Read counts were normalized by library size, and differential gene expression analysis based on a negative binomial distribution was performed using DESeq2 v3.4 (*Love et al., 2014*). In general, thresholds for differential expression were set as follows: adjusted p value<0.05, fold change >2.0 or<0.5, and average normalized read count >10. Genes were considered detectable in the immunoprecipitation samples with a normalized read

count >100. Gene set enrichment analysis was performed using GSVA v3.4 (*Hänzelmann et al., 2013*) and previously curated gene sets (*Subramanian et al., 2005*). GSEA mountain plots were generated by 'liger' R package (V0.1).

## Plasmids generation

Primers used for cloning the constructs described in the manuscript are described in *Supplementary file 6*. CD (Addgene 35102), and UPRT (Addgene 47110) were used as template for PCR for subcloning. RFP and IRES were amplified using pTRIPZ (Dharmacon) as template. The PCR products were either ligated using DNA Ligase after restriction enzyme digestion and/or by Gibson Assembly.

# Acknowledgements

We acknowledge Liping Sun and Integrated Genomics Core of MSKCC for the library preparation and sequencing of RNA-seq and Flura-seq experiments. We acknowledge MSKCC Pathology IHC lab and Marina Asher for assisting with IHC. We acknowledge MSKCC Flow Cytometry Core facility for assisting in cell sorting. We also acknowledge Andrea Ventura, Guido Wendel, Viraj Sanghvi and members of the Massague lab for intellectual discussion and critical reading of the manuscript. H.B. was supported by a Damon Runyon Postdoctoral Fellowship. KG was supported by a Conquer Cancer Foundation Young Investigator Award, an AACR Basic Cancer Research Fellowship, a Shulamit Katzman Fellowship and an ACS Postdoctoral Fellowship. This work was supported by NIH grants P01-CA094060 (JM), P30-CA008748 (JM), T32-CA009207 (KG), K08-CA230213 (KG), T32-GM07739 (YH), F30-CA203238 (YH), and a Department of Defense Innovator award W81XWH-12–0074 (JM).

# Additional information

## Competing interests

Joan Massagué: Reviewing editor, *eLife*; has filed for patent for Flura-seq method (PCT/US18/22092); serves in the scientific advisory board and owns company stocks of Scholar Rock. Harihar Basnet: Has filed for patent for Flura-seq method (PCT/US18/22092). The other authors declare that no competing interests exist.

## Funding

| Funder | Grant reference number | Author |
|---|---|---|
| National Institutes of Health | P01-CA094060 | Joan Massagué |
| Damon Runyon Cancer Research Foundation | DR-12998 | Harihar Basnet |
| Department of Defense | W81XWH-12-0074 | Joan Massagué |
| National Institutes of Health | T32-CA009207 | Karuna Ganesh |
| National Institutes of Health | T32-GM07739 | Yun-Han Huang |
| National Institutes of Health | K08-CA230213 | Karuna Ganesh |
| National Institutes of Health | F30-CA203238 | Yun-Han Huang |

The funders had no role in study design, data collection and interpretation, or the decision to submit the work for publication.

## Author contributions

Harihar Basnet, Conceptualization, Formal analysis, Funding acquisition, Investigation, Methodology, Writing—original draft; Lin Tian, Data curation, Formal analysis, Methodology; Karuna Ganesh, Methodology, Writing—review and editing; Yun-Han Huang, Data curation, Formal analysis, Writing—review and editing; Danilo G Macalinao, Data curation, Formal analysis; Edi Brogi, Resources,

Validation; Lydia WS Finley, Writing—review and editing; Joan Massagué, Supervision, Funding acquisition, Writing—review and editing

**Author ORCIDs**
Joan Massagué (iD) https://orcid.org/0000-0001-9324-8408

**Ethics**
Animal experimentation: Mouse experiments were performed following the protocols approved by the MSKCC Institutional Animal Care and Use Committee (IACUC) (#99-09-032).

**Decision letter and Author response**
Decision letter https://doi.org/10.7554/eLife.43627.036
Author response https://doi.org/10.7554/eLife.43627.037

## Additional files

**Supplementary files**
• Supplementary file 1. Genes that are differentially expressed in MDA231 cells expressing CD/UPRT and treated with indicated concentration of 5-FC for 4 hr compared to control cells were obtained by DESEQ2 analysis.
DOI: https://doi.org/10.7554/eLife.43627.014

• Supplementary file 2. Genes that are differentially expressed in TGF-β treated cells compared to SB-505124 by more than twofold are shown as identified by RNA-seq and Flura-seq. Genes commonly identified by RNA-seq 6 hr post TGF-β, but not 2.5 hr post treatment, and Flura-seq 2.5 hr post TGF-β treatment are also shown.
DOI: https://doi.org/10.7554/eLife.43627.015

• Supplementary file 3. Genes that are differentially expressed in MDA231 cells in different organs in situ as determined by Flura-seq or in vitro after isolation from the organs as determined by RNA-seq are shown.
DOI: https://doi.org/10.7554/eLife.43627.016

• Supplementary file 4. Top 100 NRF2 target genes identified by two independent ChIP-seq experiments in Hela cells (**ENCODE Project Consortium, 2012**), and the genes that were common in both experiments were used as NRF2-responsive signature genes.
DOI: https://doi.org/10.7554/eLife.43627.017

• Supplementary file 5. Genes identified to be up-regulated by more than two-fold in lung metastases compared to the corresponding primary tumors in breast cancer patients described in **Siegel et al. (2018)** for each patients are shown. Complex I genes are highlighted in red color and the total number of upregulated Complex I genes in each patient is shown.
DOI: https://doi.org/10.7554/eLife.43627.018

• Supplementary file 6. Oligonucleotide sequences used in the experiments described in the manuscript are shown.
DOI: https://doi.org/10.7554/eLife.43627.019

• Transparent reporting form
DOI: https://doi.org/10.7554/eLife.43627.020

**Data availability**
Sequencing data have been deposited in GEO under accession codes GSE93605 and GSE118937.

The following datasets were generated:

| Author(s) | Year | Dataset title | Dataset URL | Database and Identifier |
|---|---|---|---|---|
| Basnet H, Tian L, Massague J | 2018 | Organ-specific in situ transcriptomics of MDA231 cells identified by Flura-seq | https://www.ncbi.nlm.nih.gov/geo/query/acc.cgi?acc=GSE118937 | NCBI Gene Expression Omnibus, GSE118937 |

| | | | | |
|---|---|---|---|---|
| Basnet H, Macalinao DG, Massague J | 2017 | Flura-seq of TGFB treated MDA231 cells | https://www.ncbi.nlm.nih.gov/geo/query/acc.cgi?acc=GSE93605 | NCBI Gene Expression Omnibus, GSE93605 |

The following previously published datasets were used:

| Author(s) | Year | Dataset title | Dataset URL | Database and Identifier |
|---|---|---|---|---|
| Siegel M, Perou C | 2018 | Integrated RNA and DNA sequencing reveals early drivers of metastatic breast cancer | https://www.ncbi.nlm.nih.gov/geo/query/acc.cgi?acc=GSE110590 | NCBI Gene Expression Omnibus, GSE110590 |
| Minn AJ, Massague J | 2005 | ubpopulations of MDA-MB-231 and Primary Breast Cancers | https://www.ncbi.nlm.nih.gov/geo/query/acc.cgi?acc=GSE2603 | NCBI Gene Expression Omnibus, GSE2603 |
| Wang Y, Foekens J, Minn A, Massague J | 2007 | Breast cancer relapse free survival and lung metastasis free survival | https://www.ncbi.nlm.nih.gov/geo/query/acc.cgi?acc=GSE5327 | NCBI Gene Expression Omnibus, GSE5327 |
| Wang Y, Klijn JG, Zhang Y, Sieuwerts AM | 2005 | Breast cancer relapse free survival | https://www.ncbi.nlm.nih.gov/geo/query/acc.cgi?acc=GSE2034 | NCBI Gene Expression Omnibus, GSE2034 |
| Bos PD, Massague J | 2009 | Expression data from primary breast tumors | https://www.ncbi.nlm.nih.gov/geo/query/acc.cgi?acc=GSE12276 | NCBI Gene Expression Omnibus, GSE12276 |

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
