## [Decision Letter]

Thank you for submitting your article "Flura-seq identifies organ-specific metabolic adaptations during early metastatic colonization" for consideration by *eLife*. Your article has been reviewed by two peer reviewers, and the evaluation has been overseen by a Reviewing Editor and Sean Morrison as the Senior Editor. The reviewers have opted to remain anonymous.

The reviewers have discussed the reviews with one another and the Reviewing Editor has drafted this decision to help you prepare a revised submission.

The paper is interesting, though both reviewers commented that the uniqueness of the technique was overstated and raised technical questions that should be addressed in a revised manuscript. I have summarized the specific points from the reviewers that should be addressed.

Summary:

The study presents a new technique called Flura-seq for transcriptomic analysis of rare cell populations in tissues. This approach is based on 5-FU labeling of nascent RNAs in cytosine deaminase-expressing cells. After validating the approach in cell lines it was used to characterize the transcriptome of rare metastatic cells in vivo. They describe organ specific gene expression signatures and focus on the fact that lung micrometastatic cells have higher expression levels of electron transport chain as well as antioxidant genes, and confirm these findings in human breast cancer metastases.

Essential revisions:

1) With recent advances in single cell RNA sequencing, please discuss how this technique differs and, if possible, comment on how it performs in comparison to the more widely used single cell RNA sequencing approaches. A discussion of where Flura-Seq should be used instead of scRNA-seq would increase the impact of the paper.

2) Given that Flura-seq requires the overexpression of cytosine deaminase and uracil phosphoribosyl transferase, the administration of 5FU and thymine and a relatively short-term assay, these limitations should be more fully discussed. Also, the authors should not completely discount existing approaches that are relevant to these questions including laser capture microdissection/RNA-seq and SLAM-ITseq.

3) In Figure 3C, it was noted that only 53 to 74% of the aligned reads were mapped to the human genome after IP in 5-FC treated mice. Does it mean that 26 to 47% of the reads are derived from contaminating mouse reads? What does this mean for the efficacy of the staining and IP purification? Can one then trust that all the human reads correspond to stained RNA of actively transcribed genes with such a noisy technique? How good is the qPCR to assess the signal-to-noise ratio of the technique (for example Figure 3B) and why is there such a discrepancy between Figure 3B and 3C results?

4) In Figure 4D, it was claimed that lung micrometastases have the highest content of unique transcriptional activity. However, what is lacking is an objective quantification of this uniqueness. This result is the reason why the remainder of the paper focuses on the specificity of the lung versus brain metastases and electron transport chain (ETC)/antioxidant gene expression levels, so this should be more convincing.

5) Please comment on whether the main findings could be explained by the higher oxygen levels in the lung tissue compared to the brain and mammary fat pad. It is well known that high oxygen can lead to high levels of oxidative stress. This seems to be confirmed by the basal 4-HNE, GPX1 and NRF2 stainings in normal lung tissue, which are equally high as in the lung metastases and higher than in normal brain tissue (Figure 5C, E, F). Finally, in the comparison of in vitro cultures for example, the high oxygen levels could also be responsible for the loss of differences in the transcriptional profiles of ex vivo cultures from different organs (Figure 4B).

6) In co-culture experiments to assess the sensitivity of Flura-seq (Figure 1F), could the authors comment on why there isn't higher signal of human housekeeping genes in the samples when there are 10-fold more human cells?

7) Many of the RNAs downregulated in TGFb control samples are also reduced in the Flura-seq samples, which seems surprising given the short 30 minute 5-FC labeling period. Was the signal for these genes low relative to the control samples, as expected? To assess this, it would be helpful to see the normalized count values for each condition in the supplemental tables (not just fold-change). Also, are there differentially expressed transcripts identified by Flura-seq that are not found in the control samples? Are they known TGF-b targets? Potential false-positives?

8) Please clarify or reference how the 24 genes for the NRF2 signature were selected.

9) Please ensure that statistical significance of the results is addressed throughout, including in Figure 1F (where the SD is large), 2C, and 3B-C.

---

## [Author Response]

Essential revisions:1) With recent advances in single cell RNA sequencing, please discuss how this technique differs and, if possible, comment on how it performs in comparison to the more widely used single cell RNA sequencing approaches. A discussion of where Flura-Seq should be used instead of scRNA-seq would increase the impact of the paper.

We appreciate the reviewers’ suggestion. In the revised manuscript, we have added more details on the application of scRNA-seq in studying rare cells and highlighted its limitations that can be addressed by Flura-seq. Comments to this effect are included in the Introduction (third paragraph) and Discussion sections (subsection “Flura-seq as an in situ transcriptomic technique with broad biological applications”). Briefly, scRNA-seq requires disruption of tissue microenvironment to obtain single cells, and thus altering in situ transcriptiomic information. Furthermore, the low transcript coverage and the inapplicability to tissue and cell types that cannot be isolated into single cells remains a limitation of scRNA-seq.

2) Given that Flura-seq requires the overexpression of cytosine deaminase and uracil phosphoribosyl transferase, the administration of 5FU and thymine and a relatively short-term assay, these limitations should be more fully discussed. Also, the authors should not completely discount existing approaches that are relevant to these questions including laser capture microdissection/RNA-seq and SLAM-ITseq.

We agree that Flura-seq may alter certain transcripts because of CD and UPRT expression and 5-FC treatment. Therefore, it is critical to validate the discovery made by Flura-seq by alternative methods such as immunostaining, as shown in the present work. We acknowledge this caveat in the Discussion (subsection “Flura-seq as an in situ transcriptomic technique with broad biological applications”). We have discussed the utility and limitations of laser capture microdissection/RNA-seq and SLAM-ITseq in the Introduction (third paragraph).

3) In Figure 3C, it was noted that only 53 to 74% of the aligned reads were mapped to the human genome after IP in 5-FC treated mice. Does it mean that 26 to 47% of the reads are derived from contaminating mouse reads? What does this mean for the efficacy of the staining and IP purification? Can one then trust that all the human reads correspond to stained RNA of actively transcribed genes with such a noisy technique? How good is the qPCR to assess the signal-to-noise ratio of the technique (for example Figure 3B) and why is there such a discrepancy between Figure 3B and 3C results?

The remaining 26 to 47% of the reads in Figure 3C are indeed from contaminating mouse reads. The primary purpose of Flura-seq is to enrich RNAs from the cells of interest. We would like to clarify that in the experiment described in Figure 3C, the mRNAs from human cells were enriched from less than 1% in the input sample (0.003 to 0.08% of total cell numbers; Figure 3—figure supplement 1D, E) to 53 to 74%. Bioinformatic filtering based on the enrichment relative to input can remove all the reads coming from mouse sequences (Figure 3D) without significant effect on human transcript levels (Figure 3D). Such filtering is commonly used with other in situtranscriptomics methods involving immunoprecipitation or other pull down methods (Gay et al., Genes & Development 2013; Doyle et al., Cell 2008).

In Figure 3B, the signal-to-noise ratio was calculated by obtaining a ratio of the enrichment of human transcript relative to mouse transcripts that is normalized to their corresponding inputs. The data in Figure 3C is not normalized to the corresponding inputs. The normalization likely increased the ratio by 1,000 to 10,000 in Figure 3B, hence there is no discrepancy between Figure 3C and 3D. To avoid confusion, we have changed “signal-to-noise ratio” to “relative fold enrichment” throughout the manuscript including in Figure 3B, and we have clearly stated the normalization to input in the figure legends where applicable.

4) In Figure 4D, it was claimed that lung micrometastases have the highest content of unique transcriptional activity. However, what is lacking is an objective quantification of this uniqueness. This result is the reason why the remainder of the paper focuses on the specificity of the lung versus brain metastases and electron transport chain (ETC)/antioxidant gene expression levels, so this should be more convincing.

We have included a new panel to address this point (Figure 5—figure supplement 1A). We found 1528 genes upregulated only in lung micrometastases, versus 374 genes in brain micrometastases and 834 genes were commonly upregulated in both tissues compared to mammary tumor. Likewise, 1406 genes were downregulated only in lung micrometastases, versus 275 genes in brain micrometastases and 857 genes commonly downregulated in both tissues compared to mammary tumor. We hope this more clearly presents the rationale for focusing on lung versus brain metastases.

5) Please comment on whether the main findings could be explained by the higher oxygen levels in the lung tissue compared to the brain and mammary fat pad. It is well known that high oxygen can lead to high levels of oxidative stress. This seems to be confirmed by the basal 4-HNE, GPX1 and NRF2 stainings in normal lung tissue, which are equally high as in the lung metastases and higher than in normal brain tissue (Figure 5C, E, F). Finally, in the comparison of in vitro cultures for example, the high oxygen levels could also be responsible for the loss of differences in the transcriptional profiles of ex vivo cultures from different organs (Figure 4B).

It is possible that the higher oxygen concentration in lung is at least partly responsible for higher oxidative stress and the counteracting antioxidative response observed in lung micrometastases. This further emphasizes the value of Flura-seq to capture a difference that is present only in an intact tissue microenvironment. However, we cannot eliminate other microenvironmental factors in the lung as possible contributors to this transcriptional feature. We have addressed this point in the Discussion (subsection “Flura-seq as an in situ transcriptomic technique with broad biological applications”, second paragraph). It is possible that loss of differences in the transcriptional profiles of ex vivo cultures from different organs are due to high oxygen concentration in the culture condition.

6) In co-culture experiments to assess the sensitivity of Flura-seq (Figure 1F), could the authors comment on why there isn't higher signal of human housekeeping genes in the samples when there are 10-fold more human cells?

In Figure 1F, the signal-to-noise ratio was normalized to the corresponding inputs, hence the ratio remained constant between 0.1 and 0.01% human cells as the input increased with immunoprecipitated RNAs. To clarify this, we have plotted the inputs and IP separately in the new Figure 1F which shows the result as expected.

7) Many of the RNAs downregulated in TGFb control samples are also reduced in the Flura-seq samples, which seems surprising given the short 30 minute 5-FC labeling period. Was the signal for these genes low relative to the control samples, as expected? To assess this, it would be helpful to see the normalized count values for each condition in the supplemental tables (not just fold-change). Also, are there differentially expressed transcripts identified by Flura-seq that are not found in the control samples? Are they known TGF-b targets? Potential false-positives?

A closer analysis revealed that more than 90% of the repressed genes have higher reads in the Flura-seq samples compared to the control samples whereas the fold change is slightly lower in the Flura-seq samples (Supplementary file 2).This suggests that TGF-β repressed genes are regulated at the transcriptional level. This is consistent with previous reports from our lab and others that SMADs bind to the *MYC* cis-regulatory elements to repress transcription (Chen et al. Cell 2002; Frederick et al. MCB 2004).

In the revised manuscript, we have included normalized count values for all the differentially expressed genes identified in control samples by RNA-seq or Flura-seq samples in Supplementary file 2. Indeed, 575 genes were found to be differentially expressed by more than 2-fold upon TGF-β treatment by Flura-seq compared to 176 genes by RNA-seq. A majority of the genes identified only by Flura-seq show expression in the same direction in the control samples, although less than 2-fold. We have added a new figure panel (new Figure 2C) to illustrate that Flura-seq shows higher fold-change for TGF-β-induced genes, and genes that are identified only by Flura-seq are also induced in the control samples, albeit less than two-fold.

The genes identified to be induced by TGF-β only in Flura-seq are likely bona fide TGF-β target genes. To test if some of the genes in the list score upon longer TGF-β treatment, we compared the genes identified by Flura-seq to a published data set identifying TGF-β induced genes after 6 h of TGF-β treatment in MDA231 cells (Tufegdzic Vidakovic et al., Cell Reports 2015). Indeed, we found 83 of the genes were identified to be induced 6 h after TGF-β treatment (Supplementary file 2).

8) Please clarify or reference how the 24 genes for the NRF2 signature were selected.

We generated a custom NRF2 signature by using NRF2 ChIP-seq data curated by Cistrome Database. We chose ChIP-seq samples of HeLa cells, which passed all the quality controls defined by the Cistrome Database (sequence quality, mapping quality, library complexity, ChIP enrichment, signal to noise ratio, regulatory regions). The 24-gene signature was obtained by overlapping the top 100 putative targets in duplicated ChIP-seq samples, which were curated by Cistrome Database. We have cited the paper that reported the ChIP-seq data and described how the target genes were selected in the title of Supplementary file 4, and added more details in the Results section (subsection “Mitochondrial Complex I expression and oxidative stress in lung micrometastatic cells”, last paragraph).

9) Please ensure that statistical significance of the results is addressed throughout, including in Figure 1F (where the SD is large), 2C, and 3B-C.

We have added the statistical significance in all the relevant figure panels.